# Chondrocyte Hypertrophy in Osteoarthritis: Mechanistic Studies and Models for the Identification of New Therapeutic Strategies

**DOI:** 10.3390/cells11244034

**Published:** 2022-12-13

**Authors:** Shikha Chawla, Andrea Mainardi, Nilotpal Majumder, Laura Dönges, Bhupendra Kumar, Paola Occhetta, Ivan Martin, Christian Egloff, Sourabh Ghosh, Amitabha Bandyopadhyay, Andrea Barbero

**Affiliations:** 1Department of Surgery, Faculty of Medicine and Health Sciences, Research Institute of the McGill University Health Centre, Montreal, QC H3G 1A4, Canada; 2Department of Biomedicine, University Hospital Basel, University of Basel, 4031 Basel, Switzerland; 3Department of Electronics, Information and Bioengineering, Politecnico di Milano, 20133 Milano, Italy; 4Regenerative Engineering Laboratory, Department of Textile and Fibre Engineering, Indian Institute of Technology Delhi, Delhi 110016, India; 5Department of Biological Sciences and Bioengineering, Indian Institute of Technology Kanpur, Kanpur 208016, Uttar Pradesh, India; 6Department of Orthopaedic Surgery and Traumatology, University Hospital Basel, 4031 Basel, Switzerland; 7The Mehta Family Centre for Engineering in Medicine, Indian Institute of Technology Kanpur, Kanpur 208016, Uttar Pradesh, India

**Keywords:** osteoarthritis, cartilage, hypertrophy, signaling pathway

## Abstract

Articular cartilage shows limited self-healing ability owing to its low cellularity and avascularity. Untreated cartilage defects display an increased propensity to degenerate, leading to osteoarthritis (OA). During OA progression, articular chondrocytes are subjected to significant alterations in gene expression and phenotype, including a shift towards a hypertrophic-like state (with the expression of collagen type X, matrix metalloproteinases-13, and alkaline phosphatase) analogous to what eventuates during endochondral ossification. Present OA management strategies focus, however, exclusively on cartilage inflammation and degradation. A better understanding of the hypertrophic chondrocyte phenotype in OA might give new insights into its pathogenesis, suggesting potential disease-modifying therapeutic approaches. Recent developments in the field of cellular/molecular biology and tissue engineering proceeded in the direction of contrasting the onset of this hypertrophic phenotype, but knowledge gaps in the cause–effect of these processes are still present. In this review we will highlight the possible advantages and drawbacks of using this approach as a therapeutic strategy while focusing on the experimental models necessary for a better understanding of the phenomenon. Specifically, we will discuss in brief the cellular signaling pathways associated with the onset of a hypertrophic phenotype in chondrocytes during the progression of OA and will analyze in depth the advantages and disadvantages of various models that have been used to mimic it. Afterwards, we will present the strategies developed and proposed to impede chondrocyte hypertrophy and cartilage matrix mineralization/calcification. Finally, we will examine the future perspectives of OA therapeutic strategies.

## 1. Introduction

Articular cartilage has a limited capacity to self-heal owing to its avascular environment, low cellularity, and the poor proliferative activity of its resident cells, chondrocytes. Thus, untreated articular cartilage defects predispose the tissue to further degeneration leading to osteoarthritis (OA), a degenerative pathology characterized by structural, molecular, and mechanical changes in both cells and extracellular matrix (ECM) of multiple joint tissues [1,2]. Existing OA management strategies focus, however, mostly on symptoms without addressing the pathological causes. Pharmacological treatments target cartilage inflammation/degradation using non-steroidal anti-inflammatory drugs, for instance, specific cyclooxygenase-2 (COX-2) inhibitors [3] without aiming at the underlying causes. Treatments for post-traumatic lesions include restoring the joint surface after intra-articular fractures, ligament stabilization such as collateral ligament or cruciate ligament reconstruction after trauma, or biomechanical joint alignment through bone osteotomies or extra-corporal braces due to congenital malformations. More invasive strategies, finally, include subchondral bone marrow stimulation (microfracturing), autologous chondrocyte implantation (ACI), or surgical replacement of the joint [4]. Currently, therefore, no treatment option that impedes or counteracts OA-related gradual cartilage degeneration is available and there is a substantial need for a disease-modifying therapy that shields the articular cartilage from degeneration or delays OA progression.

Besides an enhanced production of matrix-degrading enzymes and a state of low-grade inflammation [5], one of the major hallmarks of OA is chondrocytes’ assumption of a hypertrophic phenotype. In OA, chondrocytes respond to the accumulation of biochemical and biomechanical injurious stimuli [6] by shifting to a hypertrophic-like state characterized by increased activity of degradative enzymes (e.g., matrix metalloproteinases-13 (MMP-13)) but also expression of markers such as collagen type X (COLX), and alkaline phosphatase (ALP) [2,7]). During the normal development and growth of skeletal elements, chondrocyte hypertrophy is a necessary transient state that culminates in bone formation through a process called endochondral ossification. In OA, however, initiation of chondrocyte differentiation into a hypertrophy-like phenotype induces a series of events that eventually cause cartilage degeneration [8]. The process begins with chondrocytes secreting degrading enzymes such as matrix metalloproteinases and a disintegrin and metalloproteinase with thrombospondin motifs (ADAMTS5) [8], and it proceeds with ECM component degradation and collagen denaturation, leading to a reduction in tissue stiffness [9]. Furthermore, there is upregulation of the hypertrophic marker COLX and Vascular Endothelial Growth Factor A (VEGF-A), which induces vascularization in the otherwise avascular cartilage [10], and of hypoxia-inducible factor 2 alpha (HIF2α) [9], a transcription factor enhancing chondrocyte terminal differentiation [11]. Finally, there is an increase in the expression of ALP [1]. Ultimately, these changes expedite patients reaching the advanced stages of OA, with extreme cartilage degradation and focal calcifications in the cartilage deep zone [12]. These changes in the calcified cartilage (i.e., the layer of mineralized tissue at the interface with subchondral bone affected by duplication of the tidemark, changes in thickness, and alteration of its mineral content [13]), and alterations in the subchondral bone (with development of new bone at the fringe between cartilage and bone, namely osteophytes, and changes including increased cortical plate thickness, subchondral trabecular bone mass and architecture variations, and enhanced bone turnover affecting its mineral content) as previously reviewed. Different works seem to even indicate that alterations in the composition of the calcified cartilage and of the subchondral bone (phenomena which are both associated with endochondral-ossification-reminiscent processes) not only contribute to further cartilage degradation but could even precede it [14,15].

In this framework, different studies have investigated the relationship between the onset of OA and articular chondrocytes’ assumption of a hypertrophic phenotype [1,6,12,16]. Rim et al. reviewed the changes that occur during chondrocyte hypertrophy or senescence in OA, highlighting how both processes are thought to play a role in OA progression but also in its initiation [1]. Reviews focusing on the relationship between chondrocyte hypertrophy and OA have also been introduced by van der Kraan et al. and Singh et al., in both cases underlining that chondrocytes lose their phenotypic stability entering a pathway to terminal differentiation. All authors agreed, however, on the fact that a direct causal relation between the onset of the chondrocyte hypertrophic phenotype and the progression of OA has not been established yet. It is still undetermined if the hypertrophy is a trigger of OA rather than the consequence of the inflammatory and degradative environment.

These considerations make the case not only for the introduction of compounds targeting hypertrophic processes as new possible disease-modifying osteoarthritis drugs (DMOADs), but also for the development of new instruments (i.e., models) to help us in dissecting which are the pathological triggers, and ultimately to possibly discover new molecular targets for promising DMOADs. Concerning DMOADs specifically, different next-generation OA treatment drugs with the aim to modify the underlining OA pathophysiology are under development and/or are in clinical trials to test their safety and efficacy, as recently reviewed by Cho et al. [17]. The majority of proposed DMOADs still target inflammatory cytokines and matrix-degrading enzymes; only a small number of compounds targeting the signaling pathway dysregulated in OA are currently in clinical trial. Among these, however, a notable candidate, lorecivivint, which acts on the wingless/integrated (Wnt) pathway involved in the development and homeostasis of joints and dysregulated in hypertrophic differentiation [18], is presently being evaluated in a phase III clinical trial in subjects with knee OA (NCT04520607). While, as specified, a comprehensive review on DMOADs under study has already been proposed we will discuss promising compounds connected with chondrocyte hypertrophic shift.

In the present manuscript, therefore, we focus both on known pathways correlated with chondrocyte hypertrophic differentiation in development and in OA, but also discuss the different in vivo and in vitro models that are available or will need to be developed to better understand these processes and ultimately possibly discover new molecular targets for promising DMOADs (Figure 1).

## 2. Chondrocyte Hypertrophy in OA

Healthy articular chondrocytes usually have a reduced proliferation capacity and do not experience terminal differentiation (leading to hypertrophy and apoptosis). Conversely, chondrocyte hypertrophy and cell death are natural phenomena during embryonic development, where chondrocytes undergo active proliferation with consequent enlargement and hypertrophic differentiation. While these chondrocytes grow significantly in size, the surrounding tissue becomes mineralized and remodels to bone via endochondral ossification [12]. This phenomenon is normal during skeletal growth, but it has also been postulated to be a pathological alteration characterizing hyaline cartilage in OA [19].

During bone development, condensation and differentiation of mesenchymal progenitor cells into chondrocytes with the expression of *SOX9* and *COL2A1* marks the first step of the endochondral ossification program. The differentiated chondrocytes then undergo transient cartilage differentiation: they start to proliferate and express Indian Hedgehog (*IHH),* and experience several steps of maturation, with the cells expressing COLX and RUNX Family Transcription Factor 2 (RUNX2), which terminally differentiate into hypertrophic chondrocytes expressing osteogenic factors such as *ALP* and Osteopontin (*OPN*). These steps are followed by invasion of the cartilage template by blood vessels from the subchondral bone and apoptosis of a large number of hypertrophic cells, eventually resulting in the cartilage template remodeling into trabecular bone [13].

Similar phenomena have been observed in OA. In fact, significant alterations ensue in articular chondrocytes during the progression of OA, both at the transcriptomics and phenotypic levels. Quiescent articular chondrocytes get activated and proliferate, and there is initiation of hypertrophic differentiation with the expression of pre-hypertrophic and hypertrophic markers such as *IHH*, *COL10A1*, and *RUNX2*, as well as production of matrix degrading enzymes like MMP-13, ADAMTS5 [2,20,21]. Moreover, OA cartilage is characterized by downregulation of articular cartilage markers such as Lubricin (*PRG4*), Aggrecan (*ACAN*), and collagen type II (*COL2A1*), owing to the augmented expression of MMPs and ADAMTS5 during OA [22,23,24]. Additionally, there is downregulation of anti-angiogenic factors such as Chondromodulin and Troponin-C that lead to increased vascularization during OA [25,26]. The endpoint is cartilage calcification and the formation of osteophytes [27], marking the advanced stages of hypertrophic OA [2]. Notably, these phenomena have been observed in human patients but could also be replicated in mouse models of OA [2,20,21]

These similarities suggest that targeting the developmental pathways activated during endochondral ossification could be a possible strategy to block OA disease progression.

Several reviews have highlighted the similarity of chondrocyte hypertrophy observed in OA and transient cartilage differentiation [8,28,29]; thus, in this article, we will only briefly discuss the resemblance of these two processes.

A list of specific markers of chondrocyte hypertrophy and cartilage mineralization, as well as the specific techniques that have been used for their characterization, are listed in Table 1.

### 2.1. Major Cellular Signaling Pathways Involved in Chondrocyte Hypertrophy, Cartilage Mineralization/Calcification and Osteophyte Formation

The signaling pathways controlling the function of chondrocytes in both permanent and transient articular cartilage are fascinating targets for disease-modifying OA therapies, and their role has already been discussed in great detail in several other reviews [8,49,50]. Among the several mechanisms that have been linked with these processes, here we focus only on those pathways that have been specifically correlated with OA-related hypertrophy (Table 2).

#### 2.1.1. IHH/PTHrP Pathway

The IHH and PTHrP signaling pathways are involved in regulating chondrocyte phenotype in the growth plate and in maintaining their homeostasis in healthy articular cartilage. IHH binds to the Patched 1 (PTCH1) receptor counteracting the inhibition caused by the Smoothened (SMO) protein and activates the downstream transcription factors GLI1 and GLI2 that translocate into the nucleus and mediate the expression of hypertrophy marker genes [29]. IHH/PTHrP signaling being a major determinant of chondrocyte hypertrophy, it was postulated to be also involved in OA pathogenesis. The parathyroid hormone (PTH) and its associated parathyroid-hormone-related protein (PTHrP) are known to function in a feedback loop with IHH to govern the progression of chondrocyte hypertrophy, linked, among others, with the adenyl cyclase/protein kinase A pathway [84].

Wei et al. were among the first to demonstrate a correlation between IHH expression levels and OA progression in patients. Comparing cartilage explants and synovial fluids from patients undergoing total knee replacement with healthy controls, the authors demonstrated that, together with the expression of MMP-13 and COLX, both the level of *IHH* gene expression in cartilage and its content in synovial fluid were significantly higher in OA patients. Moreover, MMP-13 and COLX levels in cultured chondrocytes were modulated via exogenous administration of IHH and via its knockdown through siRNA [85].

Several in vivo evaluations have also confirmed the effective role of IHH in OA manifestation through various gene manipulation studies. Zhou et al. reported a decrease in cartilage degradation and slower OA progression (with lower expression of the classic markers COLX and MMP-13) in a surgical OA mice model following cartilage-specific knockout of *IHH*. Yahara et al. reported similar results in chondrocytes specific salt-inducible kinase 3 (Sik3) [86] conditional knockout mice [87,88]. Notably, Sasagawa et al. suggested the possibility that Sik3 might be modulated by IHH and PTH/PTHrP during the progression of chondrocyte hypertrophy, reinforcing the link between OA progression and IHH modulation.

As mentioned, PTHrP is involved in a signaling loop with IHH that tightly regulates chondrocyte hypertrophy during development. For instance, mice lacking either PTHrP or its receptor exhibit dwarfism in their long bones due to premature chondrocyte maturation [89,90]. Provot et al. assessed the role of the transcriptional repressor Nkx3.2/Bapx1 in downmodulating the rate of chondrocyte maturation. Notably, Nkx3.2 expression in the limb skeleton was restricted to proliferative immature chondrocytes, and its expression in this region of the growth plate was dependent upon PTHrP signals [91]. Given this correlation between PTHrP and hypertrophy, it could be feasible to foresee a therapeutic strategy acting on Nkx3.2 to inhibit chondrocyte hypertrophy in OA. To the best of our knowledge, however, the role of Nkx3.2 has not been studied in human OA. Nevertheless, it is known that a recessive mutation of this homeobox transcription factor in human results in spondylo-megaepiphyseal-metaphyseal dysplasia (SMMD), a condition causing abnormalities in the cartilaginous growth plates of long bone [92,93].

#### 2.1.2. Wnt Pathway

Wnt is a glycoprotein secreted extracellularly whose signaling involves 19 Wnt genes and various Wnt receptors regulating both the β-catenin-dependent canonical and the β-catenin-independent non-canonical signaling pathways [94]. Wnt signaling cascades modulate biological processes that range from embryonic development and organogenesis to growth and postnatal tissue homeostasis, besides being involved in a number of diseases [18].

Regarding the relation between Wnt signaling and hypertrophy, *RUNX2*, which as we previously mentioned is one of the genes involved in chondrocyte terminal differentiation, is activated via the canonical Wnt pathway [95]. Canonical Wnt signaling occurs through the binding of one of the canonical Wnt ligands to the Frizzled family receptor and the LRP5/6 co-receptor. The signal is then transduced via GSK-3β inhibiting the β-catenin destruction complex and leading to β-catenin accumulation in the cytoplasm and its translocation into the nucleus. This causes an enhanced expression of Wnt-target genes such as *RUNX2* that further activate hypertrophic differentiation of chondrocytes through expression of COLX and ALP [29].

A pro-hypertrophic role of Wnt proteins has also been confirmed by Dong et al., albeit using chick sternal chondrocytes whose phenotype does not correlate directly with that of cartilage cells in OA patients. Specifically, the authors demonstrated that the overexpression of Wnt8c and 9c proteins causes a significant upregulation of COLX and MMP13, coupled with a strong downregulation of the chondrogenic markers SOX9 and COLII within 4 days of culture [96]. Youasa et al. detected high levels of β-catenin and its nuclear localization indicative of an active Wnt signaling in a spontaneous guinea pig OA model [97], establishing a first correlation between OA in an animal model and aberrant Wnt signaling.

Bertrand et al. were among the first to correlate dysregulated Wnt signaling with OA in human samples [98]. Specifically, the authors reported increased levels of pericellular matrix localization of Wnt3a in OA samples and that, in vitro, Wnt3a bound to basic calcium phosphate (BCP) crystals (which are highly present in the joints of OA patients) [99]. The authors suggest, therefore, a mechanism where BCP mediated sequestration of this Wnt3a positively contributes to a phenotypic shift from a healthy chondrocyte phenotype to a hypertrophic one.

In this framework, treatments that reduce the formation of BCP crystals, such as the injection of phytate (myo-inositol hexaphosphate, a natural inhibitor of calcification) might represent a good strategy to counteract chondrocyte hypertrophy in OA patients. Notably intravenous forms of phytate are indeed in early drug development stages [100].

Furthermore, van den Bosch et al. demonstrated that canonical Wnt signaling modulators such as Wnt3a and Wnt8a shifted the TGF-β signaling from ALK5 based SMAD2/3 phosphorylation to ALK1 based SMAD1/5/8 phosphorylation known to induce chondrocyte (both murine and human) hypertrophy [56]. Apart from the canonical Wnt signaling, non-canonical Wnts such as Wnt5a play a twofold role by activating chondrogenic hypertrophy during the early stage of chondrogenic differentiation through activation of G-protein coupled receptor, whereas in later stages inhibiting RUNX2 expression [55].

Finally, referring specifically to OA patients, it is worth mentioning that the mRNA expression of dickkopf 1 homolog (Xenopus laevis), i.e., *DKK1*, a Wnt signaling inhibitor, was demonstrated to be highly downregulated in the cartilage of OA patients (and with higher downregulation levels in more degraded joint areas) [101]. Moreover, DKK1 was also demonstrated to have a protective role in a mice DMM induced OA model [102] as well as in vitro with chondrocytes derived from bone marrow stem cells where DKK1 supplementation caused elevated expression of COLII and SOX9 and an equivocal decline in the expression and activity profile of COLX and ALP [103].

#### 2.1.3. TGF Beta Pathway

The TGF-β pathway has been correlated with OA and the onset of a hypertrophic phenotype in articular chondrocytes. Interestingly, however, it seems to be that the maintenance of proper levels of TGF-β is essential for chondrocyte homeostasis, whereas both excessive and inadequate activation of TGF-β signaling appears detrimental [104].

Different clinical reports have revealed an elevated magnitude of TGF-β1 in OA patients, which is instead absent or minimally present in normal articular joints [105,106]. Moreover, several experimental OA mouse models have demonstrated the contribution of the TGF-β signaling pathway in the development of osteophytes and chondrocyte hypertrophy [106,107], thus providing enough evidence about the association of the TGF-β pathway with OA hypertrophy.

A 21-day exposure to TGF-β was also found to enhance the level of hypertrophic gene expressions in a BMSCs based in vitro model [108]. This finding was further supported by the investigations from Futrega et al. indicating that a 24-h exposure of TGF-β induced stable chondrogenesis, whereas a 21-day exposure promoted hypertrophic maturation [109]. Narcisi et al. have demonstrated a loss of the chondrogenic phenotype with marked increase in COLX and IHH expression when articular chondrocytes were cultured in a serum-free medium under the presence of TGF-β [110]. Finally, a combination of in vitro and in vivo analyses by Pelttari et al. revealed ectopic hypertrophy induction during chondrogenic differentiation of MSCs under the presence of TGF-β that further instigated vascular invasion and calcification in vivo [111].

TGF-β has, however, a pleiotropic nature. For instance, it was shown that TGF-β2 could effectively inhibit the progression of OA [112,113]. Moreover, the molecule was demonstrated to promote ECM component synthesis and release [114] during joint development. This bivalent behavior was associated with the fact that TGF-β [115] can activate the chondroprotective Smad2/3 signaling route, but also aberrantly activate the pro-hypertrophic Smad 1/5/8 route contributing to the development of chondrocyte hypertrophy [108].

#### 2.1.4. BMP Pathway

Bone morphogenetic protein (BMP) plays an important role in skeletal tissue development and homeostasis. Exogenous BMP-2 and BMP-4 have been demonstrated to enhance chondrocyte differentiation of stem cells and cartilage matrix production by chondrocytes [116,117,118]. BMP-7 was evidenced to contribute to cartilage repair and suppression of cartilage degeneration [119,120]. BMP signaling is also the main driver of chondrocyte hypertrophy, eventually contributing to longitudinal bone development during the endochondral ossification processes [121]. BMP-2, highly expressed in the hypertrophic zone, specifically promotes hypertrophic differentiation of chondrocytes in the proliferative zone [122]. While cartilage hypertrophy is a necessary transient developmental stage in the growth plate, in OA it is catastrophic and initiates a cascade of events that ultimately result in permanent cartilage damage. Despite the demonstrated BMP involvement in OA-related cartilage hypertrophic differentiation, it is not well understood how each endogenous BMP and/or BMP regulator regulates this process.

BMP signaling is regulated at various levels along the signaling pathway. Among various extracellular antagonists, gene-targeting experiments have demonstrated that Noggin (NOG) and Gremlin1 (GREM1) play a role in the regulation of BMP signaling during embryonic skeletal development [123]. *NOG*-expressing cells have been demonstrated both in chick and mouse embryos to insulate transient cells from articular chondrocytes during endochondral ossification [124]. BMP signaling activity was further correlated positively with OA intensity in a collagen-induced mice model [125]. Moreover, administration of Noggin using ACLT was shown to attenuate OA traits by impeding both BMP-2 and IL-1β expression in a rat OA model [65]. In line with these in vivo results, a recent in vitro study applying a human OA chondrocyte micro-aggregate model could demonstrate a reduction in hypertrophic marker expression once BMP receptors were inhibited via LDN-193189, a small molecule BMP inhibitor [35].

GREM1, together with Frizzled-related protein (FRP) and DKK1, has also been described as a natural brake on hypertrophic differentiation in articular cartilage [126]. Leijten et al. indeed detected GREM1 presence in all zones of articular cartilage, compared to only in the resting zone in growth plate cartilage. In the same study, the authors detected a significant association between radiographic hip OA and a polymorphism (rs12593365) near the *GREM1* gene, located in a region known to regulate GREM1 expression [127]. As excessive mechanical loading has been correlated with osteoarthritis onset, GREM1 was also described as a mechanical-loading-inducible factor. In a mouse model, excessive mechanical loading was demonstrated to induce GREM1 expression via the Rac1-ROS-RelA/p65 and through activation of nuclear factor-κB (NF-κB) signaling, leading to subsequent induction of catabolic enzymes [128].

Confirming the controversial role of BMP and its regulators in OA pathophysiology, there are also studies pinpointing the application of BMP-2 protein as an anabolic pro-chondrogenic morphogen treatment for OA patients. In an Interleukin-1 (IL-1)-induced OA in vivo model, BMP-2 protein expression was correlated positively with enhanced proteoglycan turnover, suggesting association between endogenous cartilage repair mechanisms and NF-kB signaling activity [129], previously associated with BMP-2 expression in growth plate chondrocytes [130].

The role of BMP in OA is thus clearly interconnected to its cross-talk with different signaling pathways (as already described for NF-kB). Among others, the BMP and Wnt crosstalk has a central role in regulating cartilage hypertrophy, as evidenced by the presence of both BMP and Wnt antagonists (GREM1, and FRZ and DKK1, respectively) as hypertrophy brakes in articular cartilage [126]. Microarray analysis from knee joints of 15 adult human donors evidenced a higher expression of GREM1 together with WISP3 and FRZB, which modulate the Wnt-signaling pathway, in intact articular cartilage as compared to the cartilaginous layer of osteophytes. The described existence of such a “Wnt-BMP signaling feedback loop” thus poses a challenge when trying to counteract chondrocyte hypertrophy by modulating only one of these two signaling pathways.

In conclusion, BMP signaling inhibition as a target for OA treatment is still under debate and additional studies unraveling the crosstalk of BMP signaling with other OA-relevant pathways, such as NF-kB and Wnt, are needed to exclude potential side effects of BMP inhibition or BMP delivery strategies. Moreover, the high variation of currently used in vitro and in vivo OA models makes it hard to drive common conclusions. In addition, strategies targeting BMP signaling might be patient-specific, since different BMP receptor expression levels might result in varied outcomes of the treatment strategy.

#### 2.1.5. FGF Pathway

Fibroblast growth factors (FGFs) are a family of morphogens involved in processes that span embryonal development to cancer progression [131]. Raimann et al. demonstrated the production of FGF receptor 1 (FGFR1) and FGF23 by hypertrophic chondrocytes of the growth plate, connecting FGFs to endochondral ossification [132].

A first correlation between FGFs and OA was instead provided by Orfanidou et al., who identified a role of FGF23 in the regulation of RUNX2 in OA chondrocytes, which ultimately brings hypertrophy-like phenotypic changes. The authors reported also that treatment of normal chondrocytes with human recombinant FGF23 resulted in increased *RUNX2* mRNA expression [42].

A correlation between FGFs and OA was also reported by Bianchi et al. who demonstrated an upregulated expression of FGFR1 and FGF23 in OA chondrocytes in comparison to non-OA chondrocytes, with higher overexpression in more degraded areas of OA cartilage. The authors also reported an increase in hypertrophy-like changes in primary human OA chondrocytes upon exogenous exposure to FGF23, leading to an increased expression of MMP-13, COLX, and *VEGF* [133].

In contrast, Zhou et al. demonstrated that another FGF receptor, FGFR3, has a role in impeding hypertrophic phenotypic shift in mice OA-like defects, reporting an enhanced expression of hypertrophic markers *COL10A1* and *MMP13* in FGFR3 knockout mice [134]. Given their correlation with the onset of hypertrophy and the parallelism between chondrocyte terminal differentiation in endochondral ossification and OA, the above-mentioned signaling pathways might indeed present promising targets for the development of DMOADs.

A complete understanding of OA drivers has not been reached yet, however. Both arguments in favor of hypertrophy being a primary driver and a secondary event due to some other cause can be made. In this framework, in the next section we describe the different models that have been proposed to investigate hypertrophy/OA related phenomena, discussing their benefits and their disadvantages.

## 3. Models to Study Chondrocyte Hypertrophy

To discover new promising molecular targets and develop novel disease-modifying therapies, researchers have introduced various in vivo, ex vivo, and in vitro models replicating different degrees of the hypertrophic chondrocyte phenotype. The question of whether chondrocyte hypertrophy is an active or passive player can be debated by giving various views against or in support of the former or the latter opinion. Adequate OA models are therefore of paramount importance to increase our knowledge on the topic. In the subsequent section, various in vivo, ex vivo, and in vitro models related to hypertrophy and OA will be described and discussed according to their advantages and disadvantages.

### 3.1. In Vivo Models

In vivo models give arguably the most accurate and complete representation of OA and have been instrumental in understanding different mechanisms related to the pathology. They are, however, costly and it might be difficult to precisely understand the cause–effect relationship of a given phenomenon due to the presence of confounding factors intrinsic to completely biological organisms. Based on the method of induction of OA in animals, three types of models are currently used, all of which enable for the investigation of chondrocyte hypertrophy and joint tissue aberrant mineralization: (i) spontaneous models, (ii) surgically induced models, and (iii) chemically induced models.

#### 3.1.1. Spontaneous Models

Spontaneous models include mice, guinea pigs, dogs, sheep, and horses that exhibit natural OA due either to aging or genetic background [135,136,137,138]. The mice strain STR/ort develops OA at an early age and shows molecular events similar to primary OA in humans, including proteoglycan degradation, articular cartilage fibrillation, osteophyte formation, and subchondral bone remodeling [139]. Dunkin Hartley guinea pigs also develop spontaneous OA with aging with similar histopathology to human primary OA. The OA markers here appear at the age of 3 months and reach severity around 18 months [140], making them advantageous in terms of experimental timing if compared to other spontaneous OA models. However, the inactive way of living of the guinea pig limits its use in investigating the contribution of exercise to OA pathogenesis. Another gold standard model of primary OA is the canine model. OA in beagle dogs occurs after osteotomy with aging, and it shows pathological similarity to human OA, wherein the articular cartilage fibrillation appears after 7 months of tibial valgus osteotomy and the degree of severity increases with time. Eighteen months after osteotomy, the animals exhibit visible radiographic markers of OA, with marked degradation of articular cartilage and the presence of osteophytes [141]. Concerning the occurrence of spontaneous OA in horses, both idiopathic primary and post-traumatic OA have been observed [142]. Horse articular cartilage is very similar in thickness to human cartilage, with similar cellular, biochemical, and mechanical properties [143].

The molecular changes that occur in primary OA in humans and spontaneous animal models of OA seem similar, making these models, in general, particularly relevant [137]. However, spontaneous OA models have limitations such as slow progression of disease, genetic heterogeneity, the achievement of variable results, and the high associated costs that justify a rare use for clinical and translational research. As mentioned, they recapitulate a large portion of the OA phenotype in humans, including the onset of chondrocytes hypertrophic differentiation. However, as in OA patients, it is extremely difficult to determine the cause–effect relationship between chondrocyte hypertrophy and other degradative phenomena.

#### 3.1.2. Surgically Induced Models

The term surgical model refers to animals such as mice, rats, guinea pigs, sheep, dogs, and horses where OA is induced via various surgical interventions. These animal models are the most popular and widely accepted from the OA research community.

The most common surgical techniques used to induce OA in animal models are destabilization of medial meniscus (DMM), anterior cruciate ligament transection (ACLT) [135,144,145], or a combination of both if required. Transection of various other ligaments of the knee joint also induces OA with different degrees of severity in mice [146].

Small animal models are comparatively faster, cheaper, and easy to handle than large animals. However, the required surgeries need high precision with minimum collateral damage to surrounding tissues to maintain a high reproducibility, which is something more difficult to achieve in small animals.

Large animals such as dogs, sheep, and horses, are costly but present more affinity with humans in terms of joint anatomy and mechanics, thus possibly being more suitable to analyze the pathogenesis of the disease. CLT and DMM surgeries are for instance used to induce OA in large animals to mimic post-traumatic OA occurring in humans [147]. The limitations of surgically induced OA models are the need for high precision in surgery, intense post-surgical care, limited scope for screening of inflammation-related or drug-associated phenomena in the early stage of OA progression, and inability to mimic all driving forces to induce chondrocyte hypertrophy during OA progression in patients.

An example of a surgery-induced OA animal model used to investigate chondrocyte-hypertrophy-related pathways was introduced by Guo et al. [54], who studied the effects of Ipriflavone (a synthetic isoflavone derivative inhibiting IHH signaling as previously mentioned) on cartilage degeneration using an in vivo ACLT rat OA model. Ipriflavone treatment resulted in significantly less cartilage damage compared to control animals as observed from Safranin-O staining. Moreover, the expression of COLX, MMP-13, and type II collagen-C fragment was diminished, whereas COLII and ACAN expression was augmented in the Ipriflavone-treated animals. Notably, the experiments conducted in the ACLT rat model were preceded by preliminary investigations on human chondrocytes in vitro. The study could therefore be considered a paradigmatic example of how while animal models might better reflect the whole joint pathology, complementary investigations might be necessary to pinpoint the mechanism of action of a given compound in a more controlled experimental setup.

#### 3.1.3. Chemically Induced Models

Several enzymes/chemicals such as papain, sodium monoiodoacitate, and collagenase are used for the induction of OA in small and large animals. Intra-articular injection of papain in the joint cavity causes proteolytic cleavage of the proteoglycans found in articular cartilage [148]. Sodium monoiodoacetate inhibits the activity of glyceraldehyde 3 phosphate dehydrogenase, inducing chondrocyte death. As a result, massive cartilage degradation occurs as well as osteophyte formation. Collagenase induces OA by breaking down different types of collagens accountable for preserving the articular cartilage integrity, leading to hypertrophic differentiation and mineralization of articular chondrocytes [149].

Cartilage degradation through chemicals may be used to analyze pain mechanisms during OA [150]. However, the chemically induced OA has a unique pathophysiology and a limited affinity for any form of natural OA, making it a less suitable model for OA pathogenesis research in general and for the mechanisms leading to hypertrophy in particular.

Despite several animal models being developed to mimic hypertrophic OA progression, a complete recapitulation of the human OA phenotype remains elusive owing the intrinsic differences between animal and human pathophysiology [135]. Species to species variation in induction of OA may be a significant setback in yielding different OA manifestations. Moreover, animal models of OA present several drawbacks, including the faster disease progression observed in animals (different than slow clinical progression in humans) [151,152], the difficulty and high maintenance costs of large animals [135], the poor anatomical relevance of small animals [151], and ethical concerns. In this framework, the number of studies using animal models that directly addressed the onset of hypertrophy in OA is still quite limited. While animal models might be needed to confirm a certain hypothesis in a more complex environment, their usage might be better suited for later phases once a given signaling pathway or the mechanism of action of a certain drug have been clarified in a more controllable environment.

#### 3.1.4. Ex Vivo Explant Models

Ex vivo explants models refer to tissues, either articular cartilage or the whole osteochondral unit, usually in the form of biopsies or plugs, which are explanted post mortem or post-surgery (e.g., after joint replacement) and adopted in a laboratory setting. Both human and animal tissues have been used in these models, which also include whole rodent femoral heads utilized to recapitulate OA characteristics [135].

Ex vivo models offer some unique perspectives to target OA-related features, being phenotypically and genotypically similar to the in vivo counterparts, being suitable for studying inter-tissue communication, and maintaining the natural in vivo OA cartilage microenvironment. Although these explant-based OA models offer great benefits for researchers for mimicking in vivo OA microenvironments, there are challenges associated with their usage. These include the difficulty in maintaining cellular viability over a prolonged culture period limiting their ability to recapture late OA like characteristics and the requirement of high amount of material from an animal or a patient.

Different chondrocyte-hypertrophy-related studies made use of ex vivo explants. Held et al. used preserved cartilage explants from OA patients to test the small molecule inhibitors SAH-Bcl9 and StAx-35R, which block canonical Wnt signaling. The compounds were revealed to block chondrocyte phenotypic shift leading to augmented expression of *SOX9* and *ACAN* and reduced expression of *COL10A1* [59].

Sabatini et al. demonstrated that S-3429, a wide-spectrum MMP inhibitor that has a higher affinity towards MMP-13 than MMP-1, inhibited cartilage loss ex vivo using both rabbit and human cartilage fragments [153].

Tchetina et al. used full-depth human OA knee articular cartilage explants to investigate the role of TGF-β2 in suppressing the expression of hypertrophic genes such as *PTHrP, MMP-9*, *MMP-13* and *COL10A1* [154]. Guo et al. used human cartilage explants to evaluate the modulation of IHH signaling 48 h after Ipriflavone treatment. The explant culture results demonstrated that Ipriflavone blocked *RUNX2* mainly through the Smo-Gli2 pathway [54].

Venkatesan et al. utilized a human OA cartilage explant model to study the role of therapeutic gene transfer. They adopted a recombinant adeno-associated virus (rAAV) vector with a human TGF-β gene sequence for continued rAAV production of TGF-β in OA cartilage. Sustained TGF-β supply diminished the expression of COLX and MMP13. On top of providing an example of the use of an ex vivo model, the authors demonstrated an example of inhibiting chondrocyte hypertrophy as a possible therapeutic approach demonstrating how gene transfer of TGF-β might enhance cartilage-restorative events, leading to remodeling of OA human cartilage by inhibiting endochondral ossification-like events and chondrocytes terminal differentiation [155].

### 3.2. In Vitro Models

As an alternative to animal models, researchers also developed and resorted to different in vitro models to extend our understanding of hypertrophic differentiation during OA progression. These models are comprehensive of various setups ranging from simple 2D monolayer cultures to 3D in vitro models, either scaffold/biomaterial free or scaffold/biomaterial based, to models integrating mechanical load. While animal models might provide a more comprehensive recapitulation of OA phenomena [156] in general, and of chondrocyte hypertrophy in particular, in vitro models provide an easier and more profound control of the experimental environment and might serve as valuable tools to dissect OA pathogenesis, providing mechanistic insights.

The degree to which each particular in vitro model is representative of the hypertrophic phenotype of articular chondrocytes in OA patients should, however, be considered carefully. In order to draw conclusions that have the proper relevance in a clinical setting, the specific models’ limitations must be considered. In this section, we will discuss a few promising OA in vitro models that specifically aimed to replicate or account for the hypertrophic phenotype. Starting from the cellular sources of each model, our focus will be on cartilage models, while osteochondral models will not be discussed in this review.

#### 3.2.1. Cell Sources

##### Articular Chondrocytes

Articular chondrocytes (ACs), the native cell population of cartilage, are amongst the most diffused cell sources for the development of in vitro OA models. ACs are specifically adequate in studying the onset of a hypertrophic phenotype, given the unique characteristic of ACs of resisting terminal differentiation in a healthy state as previously reviewed [6].

It is important to highlight here that due to the limited availability of human cartilage donors, ACs isolated from numerous other animal species (e.g., murine, lapine, ovine, equine, porcine) have been used by researchers [157,158]. However, substantial interspecies divergences exist amongst animals and humans, e.g., cartilage thickness, proliferation rate, and propensity, as well as differentiation capacity of the isolated chondrocytes. These should be considered when using animal cells for the development of OA models [159]. Additionally, dissimilarities in terms of chondrocytes isolated from different anatomical locations such as knee and hip joints need to be considered. It is reasonably well reported that the manifestation of OA is reduced in some joints, e.g., the ankle, as compared to the knee. Eger et al. compared the cartilage from these two joints of the same limb in matched donors and demonstrated that cartilages of the two joints differed in terms of metabolic and biochemical features [160]. The expression of neoepitopes (i.e., ECM component degradation products such as NITEGE and DIPEN), for instance, was more prominent in the knee as compared to the ankle. Another interesting point to mention here is that different degenerative stages of OA can be identified concurrently within the same joint at different locations.

Osteoarthritic articular chondrocytes (OACs) show an altered phenotype compared to normal ACs, more similar to growth plate chondrocytes, that demonstrates hypertrophy-like alterations [12]. These transcriptomic differences were also demonstrated at the single-cell level [161,162].

Concerning the choice of ACs or OACs for OA models, there is not a single preferred approach but instead different choices have to be made depending on the experimental setup and hypothesis.

OACs might be used to develop a more phenotypically relevant in vitro OA model. On the other end, healthy ACs might be better suited to investigate the effect of a precise OA trigger on their phenotype without the presence of confounding factors.

Using ACs and OACs for hypertrophy-related OA models has intrinsic challenges. In vitro expansion of ACs leads to loss of their chondrogenic phenotype due to de-differentiation of the cells [163]. Expanded ACs progressively lose their differentiated phenotype already at early passages and demonstrate a pre-chondrogenic mesenchymal-like fibroblastic phenotype with the loss of proteoglycans (e.g., ACAN) and COLII synthesis and increased expression of collagen type I (COLI) [164]. To circumvent the problem of dedifferentiation when culturing ACs, Darling and colleagues suggested specific growth factor treatment during monolayer expansion or creating a more native-like expansion environment using hydrogels [165].

Additionally, OACs present problems associated with their de-differentiation during 2D culture, leading to a fast loss of OA hypertrophic traits. Moreover, only end-stage OA human cartilage is freely accessible, making reproducible sampling of OA cartilage tissue a challenging job [12]. Notably, in a previous study from our group, we demonstrated that some specific OA hypertrophic traits can be maintained in in vitro cultured OACs by using a short expansion protocol (i.e., 10 days) before culturing them in 3D aggregates under chondrogenic condition. The micro-cartilage model developed using this protocol retained the native hypertrophic OA features such as the expression of *COL10A1*, *MMP-13,* and clustered expression of COLX [35].

##### Chondro-Progenitors (ChPs)

Chondroprogenitors (ChPs) are a native cell population expressing CD105^+^/CD166^+^ or CD146^+^ and *NOTCH1* with progenitor-like behavior found in the superficial and deep articular cartilage zones that can be utilized for cartilage regeneration strategies [166].

The population has not been fully characterized, and its relatively low abundancy represents a hurdle in terms of the cell numbers required for extensive use as an OA model. The incorporation of this source in hypertrophy-related OA models, however, could shed new light on its interaction with OACs and effect on their terminal differentiation.

In the last two decades, ChPs resident in articular cartilage and their respective role in OA progression have become more prominent. Pretzel et al. demonstrated that while 15% of CD105^+^/CD166^+^ ChPs were present in normal cartilage, the percentage increased to 17% in OA [167]. Furthermore, Su et al. isolated CD146^+^ ChPs from knee joints of late-stage OA and demonstrated that these ChPs show a greater chondrogenic potential (increased *COL2A1*, *ACAN*, and *SOX9* expression) but a lower adipogenic and osteogenic differentiation potential than unsorted adipose-derived mesenchymal stromal cells and chondrocytes [168]. Koelling et al. demonstrated that ChPs isolated from end-stage OA knee joints maintained a round phenotype, akin to chondrocytes, and displayed high mRNA levels of *SOX9* and *COL2A1.* Low levels of *RUNX2* and *COL1A1* were also identified in these cells [169]. These findings were confirmed by Liu et al., who observed a significantly reduced expression of TGFβR1/ALK5 in OA-mesenchymal stromal cells compared to OA chondrocytes [170]. ChPs have been observed to have improved chondrogenic differentiation potential in comparison to bone-marrow-derived mesenchymal stromal cells and ACs [171,172] and to express low levels of COLX when cultured in monolayer [173].

##### Pluripotent Stem Cells

Embryonic stem cells (ESCs) display a high degree of proliferative capacity and pluripotency [174]; additionally, ESCs can undergo hypertrophic chondrogenic differentiation [175], highlighting their potential in the development of OA hypertrophic models. However, there are ethical and regulatory concerns related to research performed using ESCs.

Revolutionary changes have been brought in the field of tissue engineering and regenerative medicine by the introduction of induced pluripotent stem cells (iPSCs) by Yamanaka et al. [176]. The application of iPSCs for cartilage tissue engineering and OA model development has been discussed in details in a recent review by Csobonyeiova et al. [177]. It is significant to mention here that, despite many studies describing the advantages of iPSCs in cartilage regeneration and, regarding OA model development, in providing an inexhaustible supply of chondrocytes, there are several drawbacks associated with their use [178,179]. iPSCs tend to form teratomas during their differentiation, and suffer from issues such as the presence of genetic variations and the difficulty in designing culture protocols to obtain a uniform mature cell population [177]. A common disadvantage of ESCs and iPSCs is that these cells required a complex induction protocol to induce their chondrogenic differentiation.

##### Mesenchymal Stem/Stromal Cells (MSCs)

Adult mesenchymal stem/stromal cells (MSCs) [180] offer a prospective cell source option for in vitro OA model development, with several studies utilizing the multipotent differentiation capacity of MSCs to their advantage [181,182]. MSCs from different sources have been adopted or could be introduced to develop OA models representing the hypertrophic traits observed in OA chondrocytes. MSCs are, however, characterized by variations in terms of gene expression profile and differentiation capacity depending on the tissue source and the cell population and subpopulation [183]. Thus, it becomes quintessential to choose the most relevant stem cells based on the specific application. Bone marrow-MSCs (BMSCs), for instance, have the tendency to spontaneously express hypertrophy markers [182]. These cells might therefore be adopted in studying the hypertrophy-counteracting effect of promising DMOADs, but should not be used to assess the mechanisms leading to terminal differentiation in ACs.

BMSCs isolated from OA patients have been observed to express COLX, making them a promising cell source for hypertrophy mimicking in vitro OA models [184].

Adipose-derived MSCs (ASCs) are another cell source that has the ability to assume a chondrogenic phenotype [185]. While ASCs provide some additional advantages over BMSCs in terms of viability and abundance [186], there are associated drawbacks in using ASCs. A study by Mohamed-Ahmed et al. demonstrated that ASCs have reduced chondrogenic potential compared to BMSCs [187]. Moreover, in order to fully chondro-differentiate, ASCs need additional morphogens (for instance, BMP-6), thus complicating their chondrogenic differentiation protocol.

Synovium-derived MSCs were also shown to be a prospective cell source for the establishment of disease models due to their high in vitro chondrogenic capacity as compared to other mesenchymal tissues [188].

Finally, numerous studies have reported an enhanced proliferation and differentiation potential of placentally derived MSCs [189] in comparison to BMSCs [190,191]. Moreover, placental-MSCs have also been reported to express hypertrophic chondrogenic marker *COL10A1* along with the expression of *SOX9* and *COL2A1* during chondrogenic culture [150]. The umbilical cord is another exciting alternative source of MSCs [192]. MSCs isolated from Wharton’s jelly of the umbilical cord have higher proliferation and differentiation potential as compared to MSCs [193]. Moreover, they do not show any age-associated variations unlike BMSCs [194]. Human umbilical cord blood-derived MSCs (hUCB-MSCs) are another promising alternative cell source offering numerous benefits such as non-invasive collection of cells, low immunogenicity, and multi-lineage differentiation capacity [195]. A study by Young Jeong et al. found that hUCB-MSCs stimulated the differentiation of ChPs via paracrine action [196]. Nevertheless, it is important to mention that protocols for isolation and culture of fetal cells are difficult; it is also difficult to direct fetal cells towards chondrogenic differentiation, with a requirement to add additional morphogens to maintain chondrogenic culture conditions. A few other reported MSCs cell sources that offer encouraging prospects include muscle-derived MSCs [158], human nasal inferior turbinate tissue-derived MSCs (hT-MSCs) [197], and dental-pulp-derived MSCs [198].

Each cell source has, therefore, its own advantages and limitations. With regards to the elicitation of hypertrophic traits in vitro, two general considerations might be made: (i) can OA hypertrophic traits be induced in a given cell source, and (ii) can this induction be controlled?

The choice of the specific cellular source will be dependent on the answer to these questions.

Before discussing in details specific in vitro models, it is important to mention here that while quite a lot of studies are present in the literature, in the current review we have tried to focus on only those specific studies and models that mention the phenotypic change of chondrocyte hypertrophy in OA.

#### 3.2.2. Two-Dimensional (2D) In Vitro Models

Two-dimensional in vitro cultures are easy to manipulate and allow to screen several conditions simultaneously. Chondrocytes cultured in a 2D monolayer were, however, reported to dedifferentiate and spontaneously acquire a hypertrophic phenotype during the necessary expansion phases. Carol et al., for instance, highlighted how hypertrophic markers such as *RUNX2*, *COL10A1*, *MMP-13*, *VEGFA,* and *ALP* are more expressed in 2D cultures as compared to 3D models [15,199]. This behavior might mask if the assumption of hypertrophic traits is acquired because of OA like stimuli (e.g., inflammation) or if it is a bi-product of the 2D culture.

The easiest 2D OA model would consist of simply culturing OACs; however (as discussed in the previous section), there are problems associated with the de-differentiation of 2D cultured OACs leading to a fast loss of OA hypertrophic traits. Alternatively, a 2D OA model could be obtained using healthy ACs and inducing the pathological phenotype through specific inflammatory factors, such as Interleukin-1β (IL-1β) and Tumor necrosis factor-α (TNF-α) [15], or morphogenetic factors, such as TGF-β1 [200].

Different protocols have been introduced to induce hypertrophic traits in chondrocytes cultured in 2D. We demonstrated that human ACs expanded in 2D using TGF-β1 and fibroblast growth factor 2 (FGF2) were capable of differentiating toward the hypertrophic lineage (i.e., upregulating *ALP* activity in the culture medium, calcium deposition, and the expression of *COL10A1* and *BSP*) once exposed to a medium containing BMP-2 [164]. An inductive protocol based on insulin and ascorbic acid was instead reported to induce hypertrophic differentiation (i.e., COLX expression) in 2D cultured rat ACs [201]. Finally, Allas et al. developed a simple OA model using TGF-β1 treated human ACs that they used to screen EPZ6438 (tazemetostat), an inhibitor of the enhancer of zeste homolog 2 (EZH2), which counteracts TGF-β1-induced hypertrophy [107].

Caron et al. demonstrated that BMP-7 administration to both OACs and ACs treated with IL-1β or TNF-α reverted the mRNA expression levels of hypertrophic markers to those of untreated healthy ACs. Their study further reported that inducing OACs to overexpress the transcriptional repressor BAPX-1/NKX-3.2 did not change the expression of *COL2A1* and *ACAN* while decreasing the expression of hypertrophic markers such as *RUNX2*, *COL10A1, ALP* and *MMP-13* [199]. Tang et al. demonstrated that wogonoside, a specific constituent isolated from *Scutellaria baicalensis Georgi* [202], prevents IL-1β associated ECM degeneration and hypertrophic differentiation in mouse chondrocytes by inhibiting the stimulation of NF-κB/HIF-2α through the PI3K/AKT pathway [157].

Two-dimensional monolayer models are therefore diffused and have been adopted in the investigation of different mechanisms associated with hypertrophic OA traits. These methods suffer, however, from inherent limitations as these 2D models fail to replicate the 3D tissue microenvironment and physiology, thus limiting their application and predictivity as drug screening tools. Two-dimensional cultures might therefore be adopted in preliminary high-throughput screenings, but more representative models better representing the joint environment (characterized by multiple three-dimensional tissues and mechanical loading) are necessary to draw conclusions with more relevance with respect to clinical OA.

#### 3.2.3. Three-Dimensional (3D) Scaffold-Free In Vitro Models

Cell–cell and cell–extracellular matrix (ECM) interactions are fundamental constituents of the 3D tissue microenvironment; however, these conditions are not replicated in 2D monolayers, thus making the case for more complex 3D models. The simplest form of 3D models are those labeled scaffold-free. These are cellular cultures that do not include the presence of an artificial matrix used to embed the cells and can be broadly grouped in two categories: pellet cultures (i.e., small cellular aggregates) and transwell-based cultures (which might be better characterized as 2.5D models rather than 3D).

Pellet cultures represented a gold standard in tissue engineering for quite a long time. In this process, conventionally, small cellular aggregates are obtained by centrifugating the cells (cell numbers ranging from as low as 5 × 10^3^ cells [203] to as high as 0.5 × 10^6^ cells [182] per pellet) in conical tubes or in multi-well cell culture dishes. The chondrogenic medium for culturing these 3D pellets consists mostly of high-glucose DMEM containing insulin-transferrin-selenium, ascorbic acid 2-phosphate, dexamethasone, and TGF-β-1, -2, or -3 [35].

In these models, the cells can form cell–cell interactions and are free to produce their own matrix. It is therefore feasible to study which kind of ECM components are deposited as a consequence of different stimuli. De-differentiated human ACs chondrogenically cultured in 3D pellets, moreover, better recover their chondrogenic potential than in 2D cultures exhibiting increased mRNA expression of *SOX9*, *COL2A1*, and *ACAN* [204]. The cell numbers required for these 3D models, however, imply a preliminary 2D expansion phase where some unwanted unspecific hypertrophic traits are assumed by ACs. We developed a simple human 3D OA *micro-cartilage* model, using small-size cell pellets of minimally expanded OA chondrocyte that reproducibly maintained several significant characteristics of hypertrophic OA cartilage in vitro. The established *micro-cartilage* model was used to study the effect of different small molecule BMP signaling pathway inhibitors in counteracting hypertrophy-like changes in human OA articular chondrocytes [35]. A human AC pellet-based model was adopted also from Yahara et al., who reported that inhibition of the salt-inducible kinase 3 (Sik3) signaling by Pterosin B, an edible Pteridium aquilinum compound inhibited chondrocyte hypertrophy [88].

Such simple scaffold-free human OA models offer a far-reaching perspective to extend our knowledge of cartilage development and OA pathogenesis. Pellet cultures suffer, however, from intrinsic methodical drawbacks such as the necrotic core that tends to develop in the center of the pellet as a result of low diffusion and nutrients and oxygen [205]. Transwell-based OA (and more general cartilage) models are obtained by seeding chondrogenic cells in the transwell upper portion (i.e., on its porous membrane) to obtain a disc-like cartilaginous tissue [206,207].

These systems were designed to study cell–cell crosstalk. Sanchez et al. demonstrated that the transwell-based co-culture of sclerotic subchondral osteoblasts and chondrocytes leads to a hypertrophic shift and reduced expression of chondrogenic differentiation markers in ACs. These were attributed to the release of soluble growth factors such as IL-6 and TGF-β1 by sclerotic osteoblasts [196]. Murdoch et al. reported the presence of relatively high *COL10A1* mRNA levels in transwell-cultured human BMSCs in chondrogenic media. However, on comparing the chondrogenic and hypertrophic gene expression in pellet culture and transwell culture of human MSCs, they reported a weak *COL10A1* distribution at day 14 in transwells, although a strong *COL10A1* expression was observed when culturing the same cells in pellets [208]. Transwell culture models are therefore particularly suited to study possible cross talk due to factor release. Attention must, however, be paid to the fact that cells are still substantially in 2D.

#### 3.2.4. In Vitro Scaffold/Biomaterial-Based 3D Models

The monolayer and scaffold-free systems discussed until now exhibit several drawbacks such as altered cellular morphology [209], lack of mechanical stimulation [210], minimized proliferation ability [211], and poor oxygen and nutrient diffusion [212]. These challenges led scientists to develop scaffold/biomaterial-based 3D cell culture systems that consist of fibrous or porous scaffolds and/or hydrogels seeded with cells in 3D structures allowing enhanced cell–matrix and cell–cell interactions [213]. These biomaterial-based 3D models provide additional advantages such as the capability of tethering pathway-specific signaling mediators and morphogens to these matrixes, thus replicating in-vivo-like spatio-temporal gradients and the possibility of tuning their mechanical and functional properties [211,214]. With the correct choice of biomaterials and an appropriate cell source, combined with required biochemical mediators, an exact 3D microenvironment corresponding to hypertrophic OA cartilage could be mimicked, possibly dissecting the temporal evolution of the phenomena happening during OA progression [215].

Here we report different studies that made use of 3D scaffold-based models where some aspects correlated with chondrocyte hypertrophy in OA were investigated.

Tiǧli et al. [7] adopted silk and chitosan scaffolds, with and without BMP-6, to culture various cell types of human origin (BMSCs, adipose-derived MSCs, ACs, ES cells). For all these cells, COLX expression was found to be upregulated. Murab et al. developed an in vitro OA model by covalently attaching IL-1β and TNF-α to silk scaffolds, which induced higher expression of transcripts corresponding to *MMP-2, MMP-3, ADAM28,* and *ADAMTS32* in their in vitro OA model. Interestingly, higher expression levels of MMP inhibitor, TIMPs, were also observed in their OA model [216].

Nakamura. et al. used bovine-chondrocyte-loaded agarose scaffolds cultured under cyclic compression and reported an increase in the gene expression of *COL10A1* and *RUNX2* as an effect of compressive loading [217]. Dai et al. demonstrated the role of squid type II collagen (SCII) on chondrocyte hypertrophy and apoptosis. The occurrence of glycine in SCII stimulates glycine receptors present in inflammatory chondrocytes and subsides the concentration of intracellular calcium [218].

A regulatory role of Silk Fibroin-Gelatin (SF-G) in the hypertrophic differentiation of human ACs and BMSCs was observed by encapsulating cells into SF-G either in dispersed or aggregate form and fabricating the 3D bioprinted constructs. Gene expression analysis revealed a significant upregulation of *MMP-13* and *COL10A1* and a drastic decrease in *HIF1A* expression in human AC seeded SF-G construct as compared to human BMSC laden SF-G constructs [219]. To provide deeper insight into the variable role of SF-G in inhibiting chondrocyte hypertrophy, high-density murine TVA-BMSCs were seeded into the SF-G bioink in the presence of TGF-β1. An escalated level of *MMP-13*, and *COL10A1* was observed followed by a drastic decrease in chondrogenic genes (*ATX*, *PRG4*, *ACAN*) in 3D bioprinted cell-laden constructs under the influence of TGF-β1 as compared to the constructs cultured in the absence of TGF-β [108].

Three-dimensional bioprinted constructs, in particular, constitute a specific class of models where, along with the homogenous spatial distribution of cells and pores and the usage of various biomaterials and tethered molecules, it is feasible to engineer 3D tissues with a variable architecture reminiscent of their in vivo counterparts [214,220].

Despite the discussed advantage offered by 3D biomaterial-based OA models, they are also associated with certain drawbacks. For example, these models can be expensive and require specific expertise to be used, and while the wide range of biomaterials and cells available allows the replication of different OA features, additional optimization steps are required.

#### 3.2.5. Bioreactor-Based Models

OA is characterized as a complex disease with different causes [14]. A clear correlation was, however, established between OA and mechanical risk factors such as injury, obesity, and joint misalignment [221]. Moreover, mechanical stimuli have been known to regulate chondrocyte-mediated biosynthesis, remodeling, degradation, and repair of the cartilaginous tissue for more than two decades [222], and alterations in the (mechanical) properties of subchondral bone (i.e., modified trabecular structure [223,224] and tissue mineral content) are implicated in cartilage degeneration (as previously reviewed [16]).

As a consequence of these observations, different OA load-based models have been established, introducing bioreactors aimed at replicating the compressive but also the sliding motions [225,226] that characterize articular surfaces [227].

Of note, the majority of load-based models were aimed at determining how a specific mechanical stimulus influences the behavior of chondrocytes and osteochondral tissues. The focus was therefore on the altered expression, deposition, or release of anabolic markers such as GAG and COLII or the induction of inflammation and degradation. Very few of these studies centered on the assumption of chondrocytes’ hypertrophic traits. Among these, Lin et al. assessed how co-culturing (porcine) chondrocytes with osteoblasts causes a reduced expression of COLII, ACAN, SOX9, and cartilage oligomeric matrix protein (COMP), and enhanced expression of the hypertrophic markers COLX and BSP, with this shift being enhanced when osteoblasts were subjected to tensile strains [228].

Broadly, load-based models can be categorized as strain-controlled or stress-controlled (depending on the OA inducing stimulus) and scaffold/cells-based models or explant models (depending on the use of different sources). The advantages and disadvantages of these different models have been previously reviewed and will not be discussed here [156].

A particular class of (load-based) OA models are represented by those based on microfluidics-derived Organs-on-Chips (OoCs). OoCs are microscale devices aiming to recapitulate organ and tissue level functions in vitro. While OoCs have successfully been adopted in the modeling of various organs and body districts, only a few examples are present of models applying mechanical stimuli to 3D constructs [229]. Preliminary OoC based models targeting chondrocytes were pioneered by Moraes et al. [209] and then by Lee et al. [230], who proposed a device to subject 3D chondrocyte-laden hydrogel cylinders to unconfined compression but did not investigate how compression affects the assumption of an OA/hypertrophic phenotype. The first example of OA traits investigation in an OoC was proposed by Rosser et al., who reported the transient upregulation of MMP-1, MMP-3, MMP-13, and ADAMTS5 following exposure to IL-1β and TNFα [231]. This model, however, did not feature mechanical load; it was based on equine chondrocytes and was limited to short culture times. A mechanically driven OA cartilage-on-chip model was recently introduced by Occhetta et al. In this cartilage-on-chip model, OA traits such as the imbalance between anabolic and catabolic processes (e.g., a reduced expression of ACAN and a surge in the expression, release, and deposition of *MMP-13*), inflammation (i.e., an increase in the expression of *IL-6* and *IL-8*), as well as hypertrophy traits such as the increase of *COL10A1* and *IHH* expression and the decrease in the expression of genes that were demonstrated to go down with the onset of hypertrophy in OA (i.e., *FRZB*, *DKK1* and *GREM1)* could be elicited through the sole application of an hyperphysiological (i.e., 30%) confined compression [232]. Notably, the successful replication of chondrocyte hypertrophy through the sole application of mechanical stimuli could be of interest for the evaluation of disease-modifying compounds. Finally, an OoC that can apply more complex stimuli to chondrocyte constructs was recently introduced, although only applied to test the effect of different loads on cell viability [233].

Referring to chondrocytes’ hypertrophic phenotype during OA, finally, new insights will be required into the crosstalk of the cartilaginous and subchondral vascularized layers. Different OoC bone models have been proposed, as previously reviewed [234]. Most of these models were, however, aimed at studying cancer metastases rather than addressing OA related issues. While lacking the application of mechanical stimuli, more advanced milli-micrometer scale bioreactors for the study of bone–cartilage cross talks have been proposed [235,236,237] and recently reviewed [238].

While macroscale bioreactors might provide more complex stimuli (e.g., a combination of fluid-induced shear stresses, surface sliding, and compression), they are, however, bulky and might preclude their widespread diffusion given the limited experimental throughput. OoC, on the other hand, provides a more refined recapitulation of the pathological microenvironment and has the potential to be employed in high-throughput drug screenings. A compressive OA joint model on a chip where the induction of hypertrophy not only as a result of loading but also of the crosstalk, for instance, between cartilage and altered subchondral layers is still lacking, however. Despite this, the use of mechanics as an OA eliciting stimulus in vitro permits to the avoidance of possible confounding factors, making the case for further development of these models.

#### 3.2.6. Considerations on the Relevance of OA Chondrocyte Hypertrophy Models

In the chapters above, we introduced and presented different cellular models, from simple 2D cultures to more advanced, mechanically active OoCs. While drawing clinically relevant conclusions from in vitro models of chondrocyte hypertrophy, however, one should ponder (i) their relevance with respect to clinical manifestations, (ii) the specific model application, and (iii) the impossibility of decoupling these two aspects.

We saw, for instance, that while the onset of hypertrophic traits in 2D can be elicited with appropriate stimuli, some traits are the result of the non-physiological culture substrate. On the other end, even more advanced 3D pellet cultures are prone to spontaneous hypertrophic differentiation when using bone-marrow-derived MSCs. Finally, in scaffold-based approaches, different materials can induce hypertrophy traits similar to those found in vivo, but the mechanisms by which a given material induces a given phenotype are seldom explicitly explored, limiting the applicability of these models to the dissection of hypertrophic drivers in OA.

The choice, the application, and ultimately the relevance of an OA model is therefore subjective to the research question being addressed; specific models tackle specific facets of the development and progression of OA. Every model has its benefits and drawbacks and no single model offers the complete prospect to study OA. Without aiming at providing a general set of parameters for the choice of the ideal setup, three relevant examples based on our experience are presented in Figure 2. The use of 3D pellet models based on MSCs allows for studying the possible efficacy of hypertrophy-suppressing therapies and/or how the hypertrophic phenotype is modulated by common inflammatory factors found in OA (Figure 2a). The development and use of appropriate scaffolds permits the introduction and evaluation of innovate therapeutic approaches if the aim is that of targeting possible cellular therapies (Figure 2b.) Finally, an appropriate model where the inducing stimulus is highly controlled (such as in the case of mechanical stimulation) might be the more suited choice if we want to investigate the causes and the factors triggering the hypertrophic phenotype.

The importance of having OA models that represent chondrocyte hypertrophic differentiation, moreover, lies in the fact that they permit the evaluation of if a given therapeutic strategy might be efficacious in patients from a broader perspective than models, where the only OA traits recapitulated are inflammation and the onset of a degradative environment.

## 4. Strategies to Suppress Chondrocyte Hypertrophy and Cartilage Mineralization/Calcification

Having briefly presented the pathways involved in chondrocyte hypertrophy in OA and discussed different models that have been used or might be useful to better understand these processes, in this section we highlight promising clinically significant strategies aimed at targeting the onset of AC hypertrophic differentiation (Figure 2). Specifically, we discuss small-molecule-based modulation of signaling pathways, sequestration of growth factors such as VEGF, the manipulation of hypoxia, and a few biomaterials-based drug delivery systems.

### 4.1. Small-Molecule-Based Modulation of Signaling Pathways

Small molecules are compounds with a low molecular weight (e.g., in contrast, for instance, to peptides) that provide fascinating prospects as potential cell growth and differentiation factors and therapeutic agents functioning as agonists or antagonists of cellular signaling pathways [35,182,239]. These compounds are characterized by a precise chemical structure and high synthesis efficiency [239,240] and allow fast, reversible, and dose-dependent assessment of their biological response [241].

Potential DMOAD compounds (both small molecules and drugs with a higher molecular weight) have been reviewed previously [17] and will not be discussed here. We will briefly focus, however, on small molecule compounds acting on the pathways that have been correlated with chondrocyte hypertrophy in OA, specifically BMP, Wnt, and IHH/PTHrP.

Noggin is an inhibitor of the BMP signaling pathway that plays a vital role in impeding chondrocyte hypertrophy. Experimental evidence has revealed that Noggin successfully downregulated IL-1β and BMP-2 mediated chondrocyte hypertrophy in murine chondrocytes [65]. Another BMP signaling inhibitor is constituted by chordin, which was demonstrated to contrast chondrocyte mineralization and ALP activity via competitive binding mechanisms [242,243]. Dorsomorphin was the first identified small molecule antagonist of the BMP signaling cascade that operates via the inhibition of BMPR1 thereby inhibiting the BMP-activated Smad 1/5/8 phosphorylation, thereby blocking chondrocyte hypertrophy as measured by ALP activity [244]. Chawla et al. also demonstrated the hypertrophy inhibitory potential of LDN193189, a small molecule specific inhibitor of BMP signaling that induced downregulation of *COL10A1*, *MMP-13*, and *IHH* gene expression in human OACs [35].

Caron et al. identified many overlapping oligopeptide sequences (p [63–82] and p [113–132]) from the BMP7 peptide library that, even with a single dose, modulated *COL10A1* and *MMP-13* expression in OACs by altering the SMAD signaling [245].

As previously reviewed, a second promising target for osteoarthritis therapy connected with the onset of a hypertrophic phenotype is constituted by Wnt signaling [95].

As an example, the Wnt signaling pathway inhibitor PKF118-130 administered to MSCs directed towards chondrogenic differentiation induced upregulation in chondrogenic gene expression and a marked downregulation of hypertrophic markers [57].

On a different note, in an in vitro chondrocyte model, the cyclooxygenase-2 (COX2) inhibitor Rofecoxib (even though COX2 is not a direct Wnt inhibitor) reduced the expression of hypertrophic markers (COLX, RUNX2, ALP, PGE-2) while also downregulating the expression of different downstream targets of the Wnt cascade (i.e., β-catenin, Axin 2, and GSK3β) [246].

Most notably, moreover, a small-molecule inhibitor of the Wnt signaling pathway, SM04690, (Lorecivivint) is being evaluated as a possible DMOAD in a phase III clinical trial (NCT04520607) after the success of phase I and II [247]. Lorecivivint causes inhibition of CDC-like kinase enzymes (CLK2) leading to reduced expression of Wnt target genes and inflammatory cytokines [248].

In addition, two IHH inhibitors have been proposed as therapeutic candidates: Cyclopamine and Ipriflavone. Cyclopamine is a IHH inhibitor demonstrated to downregulate the expression of hypertrophy hallmark genes *COL10A1* and *MMP-13*, thereby mitigating OA progression in mice [87]. Ipriflavone is a dietary supplement shown to efficiently inhibit IHH signaling in both mouse and human cells without cytotoxicity (and to be 10-fold safer than cyclopamine), reducing the expression levels of IHH signal pathway genes (such as *SMO*, *Gli2*, *RUNX2*) [54].

Finally, another small molecule, G141, a fibroblast growth factor receptor-I (FGFR-I) inhibitor, has shown promising results in mitigating hypertrophy and OA. Knockdown of FGFR-I led to a diminished expression of hypertrophic marker genes in a murine OA model, substantiating the role of FGFR-I in chondrocyte hypertrophy [83]. Chondrocytes primed G141 and demonstrated lower mRNA levels of *MMP-13* and *ADAMTS5,* as well as a partial restoration of chondrogenic markers *ACAN* and *COL2A1* [232].

Small molecules have, therefore, undeniable advantages and are showing promising results. The cellular signaling pathways that constitute the target of these compounds are, however, under strict spatio-temporal control during the development and homeostasis of articular cartilage and possibly act in multiple ways. There is, therefore, a stringent requirement to delivery modality for small molecule inhibitors in vivo in a localized manner (i.e., through intra-articular injections) in order to evade possible off-target toxicity.

### 4.2. Role of VEGF Sequestration and Hypoxia

The similarity between endochondral ossification and molecular changes witnessed in OA hint towards the involvement of angiogenic and hypoxic mediators such as VEGF and HIFs in the pathophysiology of OA.

For example, a correlation has been established by genome-wide association studies between enhanced expression of VEGF and the extent of cartilage degeneration in OA [249], leading to the notion that sequestering microenvironmental VEGF as a method of angiogenesis blockade might function as a promising strategy to suppress OA associated hypertrophy.

Marsano et al. retrovirally transduced BMSCs to express a decoy soluble vascular endothelial growth factor (VEGF) receptor-2 (sFlk1), which efficiently sequesters endogenous VEGF in vivo, seeded them on collagen sponges, and studied their chondrogenic potential in vitro and in vivo after ectopic implantation in nude mice. sFlk-1-BMSCs inhibited angiogenesis in both in vitro and in vivo conditions and developed phenotypically stable cartilage with no evidence of expression of COLX, MMP-13, and BSP [250]. The same group also examined the effects of a microenvironment blockade of VEGF on the chondrogenic potential of human nasal chondrocytes via two strategies, using the sequestration of VEGF through sFlk-1 or using anti-angiogenic hyperbranched peptides. Independently from the blockade strategy, VEGF inhibition strongly improved in vivo cartilage’s ability to form nasal chondrocytes, as demonstrated by enhanced GAG deposition [251].

Cartilage is characterized by a hypoxic microenvironment. The role of hypoxia in suppressing OAC hypertrophic characteristics and the molecular mechanism of action of hypoxia have already been discussed quite well in several other reviews [29,50,252]. Here, we will discuss only a few studies describing hypoxia as a factor to suppress chondrogenic hypertrophy in OA.

Allas et al. reported the role of dimethyloxalylglycine (DMOG), a hypoxia-inducible factor prolyl hydroxylase, in simulating hypoxia-like stabilization of HIF factors while improving their nuclear localization. DMOG reduced the expression of COLX and MMP-13 in a TGF-β1 induced in vitro OA model [107].

Markway et al. demonstrated in a pellet culture OA chondrocytes model that in comparison to normoxic conditions (i.e., 20% oxygen (O_2_)), cells cultured under 2% hypoxia (2% O_2_) downregulated the expression of the hypertrophic markers *COL10A1*, *MMP-1,-2,-3,* and *-13* while maintaining the expression of *COL2A1* and *ACAN,* demonstrating that hypoxia plays a role in the suppression of hypertrophic characteristics in OA chondrocytes [253]. Marsano et al. further demonstrated that hypoxia (2% O_2_) induced chondrogenesis and suppressed hypertrophy in pellet cultures of human BMSCs even without any chondrogenic differentiation factor [250]. Thus, maintaining a low oxygen concentration that mimics the native cartilage microenvironment might be a prospective strategy to limit or suppress hypertrophy in OA cartilage. Szojka et al. further demonstrated the role of mechano-hypoxia conditioning in suppressing hypertrophy during chondrogenic differentiation of meniscus fibrochondrocytes and highlighted the significance of mechano-hypoxia in inducing non-hypertrophic articular cartilage in MSCs [254]. Based on this evidence, targeting VEGF and hypoxia could be helpful to develop therapeutic strategies targeted at downregulating MMP production and/or suppressing hypertrophy in OA, or even for and the development of healthy cartilage tissue.

Further investigations will, however, be needed to properly assess, for instance, the differences between using these strategies to drive chondrogenic differentiation in MSCs and to prevent or revert the hypertrophic phenotype in differentiated articular chondrocytes.

### 4.3. Biomaterial-Based Drug Delivery Systems to Mitigate Chondrocyte Hypertrophy

A wide range of administration routes (e.g., oral, intravenous, intramuscular) have been proposed to deliver various anti OA drugs to the desired region, but achieving the necessary local intra-articular concentration while avoiding systemic adverse effects remains challenging [255,256]. Furthermore, even intra-articular administration is affected by rapid clearance of small and macro-sized molecules by synovial microvasculature [257]. Biomaterial-based drug delivery systems are 3D constructs based largely on hydrophilic polymers that can be used to ameliorate the administration of a given drug.

As previously extensively reviewed [258], biomaterial-based drug delivery systems can increase the shelf-life and retention time of a given compound while also allowing sustained release of the loaded molecule in time with a more physiological concentration profile so as to ensure a long-term therapeutic outcome with minimized adverse effects.

Moreover, biomaterials such as chondroitin sulfate or hyaluronic acid have been shown to downregulate hypertrophy by themselves via inhibition of the Smad 1/5/8 pathway and upregulation of the p38 MAPK pathway [259,260]. Such biomaterials, possibly combined with 3D bioprinting approaches, could be used to develop anatomically relevant OA cartilage tissue equivalents to be used in repair strategies.

In this paragraph we will describe a series of polymeric materials that, in combination with different bioactive compounds, has been suggested for combating chondrocyte hypertrophy in OA.

TGF-β3 loaded alginate microspheres, when incorporated in hyaluronic acid (HA) hydrogel discs together with hBMSCs and primed in chondrogenic medium, demonstrated a controlled release profile leading to a significant upregulation of chondrogenic gene expression. Moreover, implantation of the TGF-β3 loaded hydrogel discs into the subcutaneous pocket of nude mice led to a further increase of proteoglycan and collagen content, highlighting the chondroprotective role of the system with respect to hypertrophic maturation [261]. Another study by Bello et al. used ASC spheroids co-loaded with TGF-β3/Matrillin-3 laden gelatin microparticles as a potential carrier to suppress chondrocyte hypertrophy. The authors observed a spike in the gene expression of cartilage-specific markers (i.e., *COL2A1*, *ACAN*) with a high decrease in the expression of hypertrophic markers (i.e., *MMP-13*, *RUNX2*) [262]. Such efficient inhibition of the hypertrophic differentiation (also validated by an in vivo rat model) could be attributed to the synergistic inhibitory potential of TGF-β3 via the Smad pathway and Matrillin-3 via the BMP-2 inhibitory mechanism [64,263].

Similar effects were also obtained with HA conjugated with epigallocatechin-3-gallate gelatin hydrogel encapsulated with porcine ACs and primed in chondrogenic differentiation medium, which led to a significant downregulation of hypertrophic markers *MMP-13*, *IL-1β*, and *ADAMTS5* [264].

Platelet-rich plasma (PRP) is known to exhibit anti-inflammatory potential leading to a reduced expression of MMPs and disintegrins in human OACs by inhibiting the IL-1β mediated NF-κB signaling pathway [265,266]. Therefore, rat chondrocytes encapsulated in tyramine-conjugated gelatin-PEG hydrogels loaded with PRP exhibited a marked increase in the expression of chondrogenic lineage markers (e.g *ACAN*, *SOX9*, *COL2A1*) [267]. However, a fascinating observation from the experiment lies in the upregulated expression of anti-angiogenic marker (ChM1) and matrix degradation inhibition marker (CB1), indicating a possible role of PRP in attenuating chondrocyte hypertrophy by inhibiting neo-angiogenesis and matrix degradation.

Hou et al. investigated the therapeutic effect of Kartogenin (KGN), a small molecule regulator of chondrogenesis, and demonstrated a significant decrease in the expression of *MMP-13* and *ADAMTS5*, an enhanced cartilage ECM deposition, and an increase in the expression of *COL2A1* and *ACAN* in KGN supplemented chondrocytes [268]. KGN was furthermore adopted, together with a fibrin scaffold, in a comparative analysis with respect to TGF-β3 that revealed the superior reduction in *COL10A1* expression from hASCs using KGN [269].

## 5. Conclusions and Future Perspectives

OA is a complex, multifactorial disease correlated with multiple environmental and genetic factors that not only leads to degradation of the articular cartilage but also affects menisci, subchondral bone, and the synovial membrane, as well as periarticular muscles and ligaments.

Furthermore, while the precise pathogenic mechanism of OA has not yet been uncovered, it is expected that cartilage degeneration and joint function loss occur by more than one means, which can further vary for distinct OA subgroups depending on their pathogenesis (e.g., age-related or post-traumatic OA). These observations render the individuation of a single strategy to counteract OA extremely difficult to pinpoint.

One of the most accepted routes of pathological progression, however, highlights how ACs in OA are subjected to a phenotypical instability leading to differentiation processes similar to those observed in chondrocytes during endochondral ossification. Cartilage wear and tear at the joint surface leads to expansion of the calcified cartilage zone as cartilage fissures/cracks propagate. In an attempt to repair the tissue, chondrocytes show enhanced proliferation and synthetic activity, but in doing so produce matrix degradation products and proinflammatory cytokines. These factors disrupt the normal chondrocytes’ function and quiescent state while also establishing a crosstalk with the synovium that ultimately enhances proliferative and pro-inflammatory responses.

In this review we not only focused on the mechanisms leading to the onset of a hypertrophic phenotype in chondrocytes in OA, but also presented and discussed both in vitro and in vivo models that have been used or might be used to increase our knowledge of these processes.

Notably, we only focused on cartilage models, but more complex setups also involving synovium and subchondral bone are needed to properly understand the complex intra-tissues crosstalk that characterizes OA.

In the present state, it is not known whether inflammation and cartilage degradation in general follow hypertrophy or induce it [270,271]. Interestingly, however, inhibitors of chondrocyte hypertrophy have been shown to suppress the expression of inflammation markers [272]; as mentioned, lorecivivnt, a Wnt inhibitor, reached a phase III clinical trial as a DMOAD, indicating that acting on the pathways leading to hypertrophy seems to be an effective strategy to counteract the progression of OA.

For instance, it is known that following immobilization of rodent joints, hypertrophy sets in in the articular cartilage, but remobilization leads to the significant recovery of joint architecture and cellular nature [273]. Thus, the question is, do the same cartilage cells that became hypertrophic following immobilization recover their phenotype and go back to expressing COLII, or do the hypertrophic cells die and are replaced by new COLII expressing cartilage cells? With the reversal of hypertrophy in chondrocytes via small molecules or biomaterials modulating signaling pathways, can the associated molecular changes provide us further insights into possible gene targets for disease modifying therapy of OA?

Additional studies will be needed to answer to these and other questions. Presenting a summary of the dysregulated pathways and of the models used to investigate them, we tried to deliver a basic compendium that could be adopted to instruct possible new research and the development of more complex models.

## Figures and Tables

**Figure 1 cells-11-04034-f001:**
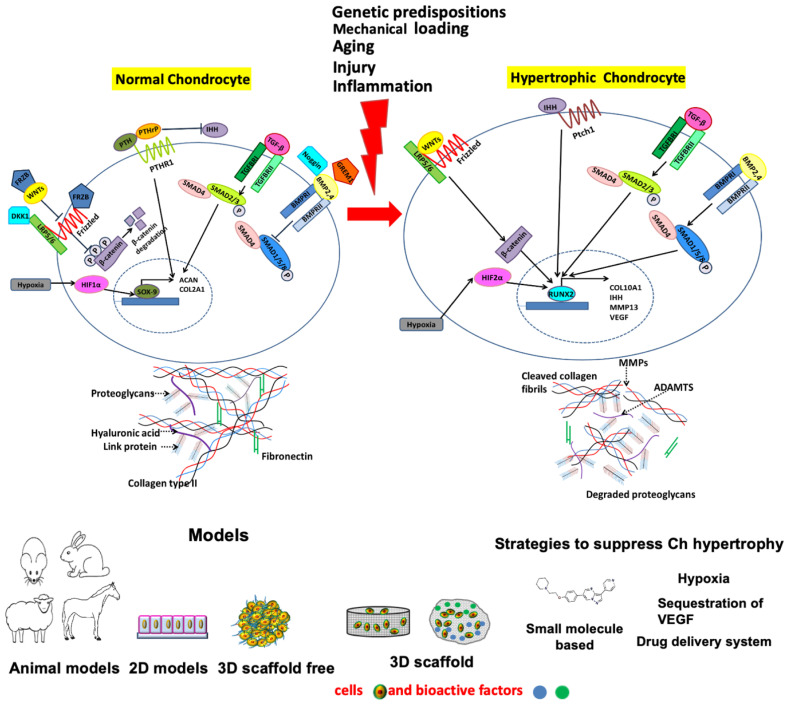
Schematic describing the review summary; the differences in the normal and hypertrophic chondrocyte in terms of active cellular signaling pathways corresponding to healthy ECM and degraded ECM, respectively; models that have been utilized to simulate OA specific hypertrophy; and the basis of the strategies to suppress hypertrophy.

**Figure 2 cells-11-04034-f002:**
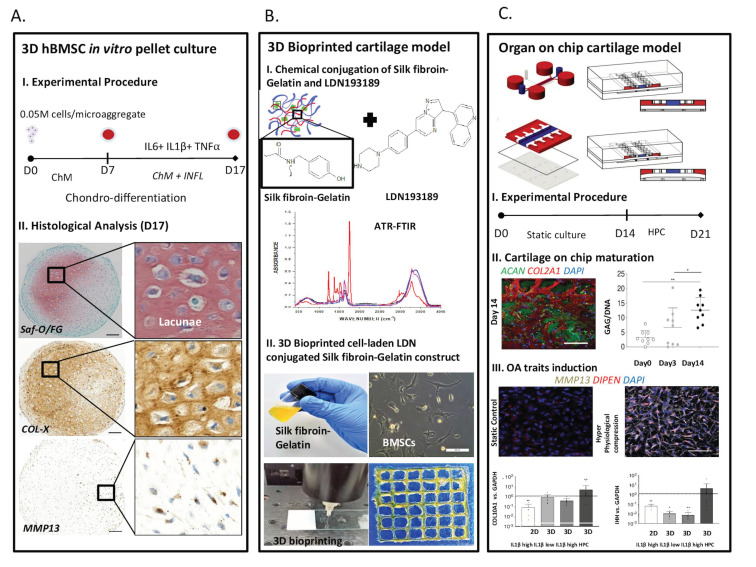
Strategies to recapitulate OA chondrocyte hypertrophy in possibly clinically relevant in vitro models to study the triggers leading to the phenotype and evaluate possible OA reversing treatments. (**A**) hBMSCs were chondro-differentiated in 3D microaggregates using chondrogenic medium (ChM) for 7 days. Chondrogenic pellets were subsequently exposed to low-grade inflammatory factors (INFL; i.e., IL6, IL1β, TNF⍺) present in the OA joint. At the end of such OA-mimicking culture phase, characteristic OA features such as lacunae formation, COLX, and Matrix MMP-13 expression were detected. (**B**) Development of a 3D bioprinted phenotypically stable articular cartilage model. The BMP cascade inhibitor LDN193189 is chemically conjugated with SF-G bioink via a cyanuric chloride conjugation reaction. The chemical attachment is further confirmed through ATR-FTIR spectroscopy. The LDN193189-laden SF-G blend is encapsulated with BMSCs and enzymatically crosslinked with mushroom tyrosinase followed by 3D bioprinting using a direct write assembly. Successful chondrogenesis is achieved through simultaneous inhibition of the BMP signaling pathway by LDN193189 and upregulation of the WnT signaling pathway through SFG, confirmed via gene expression and protein expression analysis. (**C**) Mechanically driven OA cartilage on-chip model (adapted from Occhetta, Mainardi, et al.). A mechanically active OoC model is obtained by introducing two compartmentalized PDMS microchambers separated by a PDMS membrane (left). The top compartment is subdivided, by two rows of hanging posts (in white), into a central channel hosting the 3D microconstruct (in blue) and two side channels for medium supplementation (in red). The bottom chamber (in grey) represents the actuation compartment. By pressurizing the bottom compartment, the PDMS membrane deforms, eventually abutting against the posts’ ends and causing confined compression of the microconstruct (bottom right). After 14 days of differentiation, there is cartilage-like ECM deposition (left, scale bar 100 μm) and quantifying GAG and DNA (right). After 21 days of hyperphysiological compression, an OA phenotype was elicited, as demonstrated by the increased expression of MMP-13 and DIPEN (top, scale bar, 100 μm). Moreover, the hyperphysiological compression triggers a hypertrophic phenotype with an increased expression of *COL10A1* and *IHH*, different from 2D and 3D cytokine-based models, where these markers are diminished with respect to non-stimulated controls.

**Table 1 cells-11-04034-t001:** Analyses to assess the extent of chondrocyte hypertrophy and cartilage mineralization.

Type of Analyses	Marker	Technique
Composition of the extracellular matrix	Glycosaminoglycans (GAG)	Histology (Safranin-O staining) Biochemistry (DMMB GAG quantification) [30]
	Calcium (Ca) deposit	Histology (Alizarin red staining) Biochemistry (Ca quantification) [31]
	COLX *COL10A1*	Immunostaining Western blot In-situ hybridisation [32,33]
	MMP-1, -2, -13 *MMP1, MMP2, MMP13*	Immunostaining Western blot qRT-PCR [34,35]
	MMP-derived fragment of type II collagen	Histology [34,35]
	Biochemical components	RAMAN spectroscopy [36,37]
Properties of the tissue and or ECM	Collagen fiber breakdown	Scanning electron microscopy [38] Histology Transmission electron microscopy
	Matrix stiffness	Atomic Force Microscopy [39]
Assessments of (sub)cellular properties	Hypertrophic markers (*ALP*, *IHH*)	qRT-PCR [40]
	(Frizzled-related Protein (*FRZB*), Gremlin (*GREM1*), Dickkopf-I, (*DKK1*) (decreased expression in OA)	qRT-PCR [41]
	*RUNX2* (total) RUNX2 (nuclear)	qRT-PCR Western blot [42]
	Phosphorylated Smad1, Smad5, and Smad9	Immunostaining Western blot [43]
	Bone morphogenetic protein type I receptors, Activin A receptor like type 2,3,6 (ALK2,3,6)	Immunostaining Western blot [35,44]
	Transglutaminase 2 (TG2)	Immunostaining Western blot [45]
Analyses of the degradome and the extracellular vesicles	Ca released	Biochemical quantification [46]
	ALP activity	Biochemical quantification [46]
	MMPs activity	Biochemical quantification Zymography [47]
	MMP-derived fragment of type II collagen and aggrecan	Immunoassay [48]

**Table 2 cells-11-04034-t002:** Signaling pathways and their modulators involved during hypertrophic OA progression.

Signaling Pathway	Role in OA Progression	Pathway Specific Inhibitor	Mechanism of Action of Inhibitor/Activator
IHH/parathyroid hormone-related protein (PTHrP) signaling	IHH signaling activates OA hypertrophy aided by *RUNX2*. PTHrP selectively inhibits hypertrophy by acting in a negative feedback loop with IHH [51,52].	HDAC4 (Inhibitor)	Downregulates *RUNX2* expression and thus regulates the IHH signaling pathway [53].
		Ipriflavone (Inhibitor)	Blocks IHH pathway [54].
WNT signaling	Binding of Frizzled receptor and low-density lipoprotein receptor-related protein (LRP) 5/6 to WNT ligand enhances nuclear translocation of Beta-Catenin (β-catenin) and causes the expression of *RUNX2*, further initiating hypertrophy [29]. Non-canonical Wnts (e.g., Wnt5A) play a dual role. Wnt5A activates hypertrophy during the initial stages of chondrogenic differentiation, while in later stages inhibits *RUNX2* expression [55,56].	DKK1 (Inhibitor)	Interacts with low-density lipoprotein receptor proteins (LRP-5 and LRP-6), and inhibits the formation of the WNT-Fz-LRP complex [41].
		FRZB (Inhibitor)	Develops a non-functional complex with Frizzled receptors inhibiting WNT/β-catenin signaling [41].
		EPZ005687 (Inhibitor)	Inhibits enhancer of zeste homolog 2 (EZH2), a histone methyltransferase that is involved in the induction of hypertrophic OA, by blocking WNT/β-catenin signaling [57].
		PKF118-130 (Inhibitor)	Inhibits WNT signaling by inhibiting nuclear translocation of β-catenin, thus enhancing the chondrogenic marker expression, while reducing the expression of hypertrophic markers [58].
		A stapled peptide derived from the Bcl9 homology domain-2) (SAH-Bcl9), Stapled β -catenin binding domain of Axin (StAx-35R) (Inhibitor)	These small molecule inhibitors inhibit canonical WNT signaling, thereby inhibiting hypertrophic chondrocyte shift, increasing the gene expression of *SOX9* and *ACAN*, and decreasing the expression of *COL10A1* [59].
Transforming growth factor-β (TGF-β) signaling	High TGF-β1 levels have been observed in OA patients leading to osteophyte development and chondrocyte hypertrophy [60,61].	SB505124 (Inhibitor)	Blocks TGF-β type I receptor, thus inhibiting TGF-β activity and reducing the degeneration of OA articular cartilage [62].
BMP signaling	Increased phosphorylation of intracellular SMAD proteins (SMAD1/5/8) leads to enhanced nuclear translocation of SMAD4, inducing hypertrophy. Increased BMP-2 protein expression has been detected in human OA cartilage [8,63].	LDN193189 (Inhibitor)	Blocks BMP signaling by selective inhibition of ALK2/3 and suppresses hypertrophic OA traits, thereby reducing the expression of COLX and MMP-13 [35].
		Matrilin-3 (Inhibitor)	Inhibits binding of BMP-2 with its receptor by interacting with BMP-2 ligand, thus inhibiting downstream BMP signaling and decreasing the hypertrophic marker COLX [64].
		Noggin (Inhibitor)	Blocks BMP-2 activity by inhibiting the binding of BMP-2 with its receptors, reducing cartilage degradation in OA [65].
		SMAD7 (Inhibitor)	Inhibits Smad pathways in chondrocytes in vivo. Smad7 deficiency leads to a reduction in the hypertrophic zone [66].
Calcium signaling	A rise in extracellular calcium and increased activity of calcium-sensing receptors have been linked to COLX up-regulation during OA. The binding of calcium to calmodulin activates Calcium/calmodulin-dependent protein kinase, inducing hypertrophy [67,68,69].	-	-
Integrin signaling	Overexpression of integrin pathway modulators RhoA/Rock suppresses ALP and mineralisation in chondrocytes [70,71,72].	Function-blocking anti-integrin β1 antibody (Inhibitor)	Inhibits COLX expression and hypertrophy [73].
Notch signaling	Enhanced mRNA expression of Notch ligand *Jagged 1* and its receptor *Notch 1* in human OAACs has been identified [51,74].	N- [N-(3,5-diflurophenylacetate)-L-alanyl]-(S)-phenylglycine t-butyl ester (DAPT). (Inhibitor)	Intra-articular injection of DAPT in mouse knee reduced hypertrophic OA progression [51,74].
MAPK pathway (p38, c-Jun N-terminal (JNK) kinase, and extra-cellular-regulated kinases (ERK))	MAP kinases act as key mediators that regulate the expression of MMPs during OA. Activation of p38 represses COLX expression and OA progression. Phosphorylation of ERK1/2 increases in OA with an increased hypertrophic phenotype [75,76,77].	U0126 (Inhibitor) Sprouty RTK signaling antagonist 4 (Inhibitor MAPK)	Inhibits the MEK-ERK pathway leading to reduced pERK levels and diminished expression of *RUNX2,* *COL10A1*, *ADAMTS5,* and *MMP-13* [77]. Inhibits MAPK pathway leading to inhibiton of chondrocyte hypertrophy [78].
AMP-activated protein kinase (AMPK)/PI3K/AKT signaling pathway	Reduced AMPK and PI3K-AKT expression have been observed in OA articular cartilage [79,80,81].	Asiatic acid (Inhibitor PI3K/AKT) (Activator)	Inhibits the phosphorylation status of PI3K/AKT and activates the phosphorylation of AMK, contributing to the reduction in hypertrophy [82].
FGF signaling	Enhanced FGF23 and FGF1 in OA chondrocytes [29].	G141 (Inhibitor)	Reduces the expression of hypertrophic markers *MMP-13* and *COL10A1* and reduces cartilage degradation [83].

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
