# Peer review of "Chondrocyte Hypertrophy in Osteoarthritis: Mechanistic Studies and Models for the Identification of New Therapeutic Strategies"

_cells, 2022, doi:10.3390/cells11244034_

Round 1

Reviewer 1 Report (Previous Reviewer 3)

This twice- revised manuscript is much improved according to the comments of reviewer 1. I do not have any additional comments. 

Reviewer 2 Report (Previous Reviewer 1)

The quality of the manuscript has much improved. I have no further comments.

This manuscript is a resubmission of an earlier submission. The following is a list of the peer review reports and author responses from that submission.

Round 1

Reviewer 1 Report

The manuscript by Chawla et al. is a try to review whether suppression of chondrocyte hypertrophy could be a promising therapeutic strategy for osteoarthritis. This is a large and evolving field with pharmacological interventions that have approached as far as into phase III clinical testing. From a review with this topic and title, I would have hoped to get answers to the following basic and important questions: what is the convincing evidence that chondrocyte hypertrophy is an active driver rather than a passive consequence of OA? To what degree is the chondrocyte hypertrophy observed during OA similar to that during embryonic bone development and in the vast amount of in vitro cultures of hypertrophic chondrocytes? Are there any parameters to judge whether a given model is a suitable model of OA chondrocyte hypertrophy? To what degree are results from any of the models translatable to OA? 

Unfortunately, the manuscript in its current form is a rather loose collection of various topics around chondrocyte hypertrophy, while the OA focus is often lost. The manuscript would very much benefit from a more stringent focus on OA and a higher degree of abstraction.

Specific comments:

1.     Pro-hypertrophic signaling pathways: The length of description appears to be biased by the group’s research focus on BMP signaling. PTHrP/IHH and WNT signaling should be described in similar depth. For each pathway: have interventions been tested for their potential as DMOADs?

2.     Models: please evaluate each model not only regarding their technical challenges but more importantly regarding their potential to model OA and translatability of results. What is wrong with our models, what should we change, given the alarmingly high rate of initially promising ‘DMOADs’ that turn out to be non-effective?

3.     Chapter Strategies to suppress chondrocyte hypertrophy: please more strongly focus on the hypertrophy aspect. Only one of the three strategies depicted in figure 2 directly targets hypertrophy (via BMP inhibition), while the others target other typical OA drivers: inflammation and mechanical overload. For each potential intervention (small molecules, VEGF sequestration, hypoxia, biomaterials): how can the described insights be harnessed to potentially treat OA?

Author Response

Reviewer 1

The manuscript by Chawla et al. is a try to review whether suppression of chondrocyte hypertrophy could be a promising therapeutic strategy for osteoarthritis. This is a large and evolving field with pharmacological interventions that have approached as far as into phase III clinical testing. From a review with this topic and title, I would have hoped to get answers to the following basic and important questions:

  1. what is the convincing evidence that chondrocyte hypertrophy is an active driver rather than a passive consequence of OA?

Response: We agree with the Reviewer on the fact that this answer was not given in our text. Given the present state of the literature, howeverer, there might be not enough evidence to determine if the case is the former or the latter.

We modified the text of the manuscript in multiple points to highlight that (i) it is presently not clear if the hypertrophic state of chondrocytes in OA is a causative phenomenon or the results of other triggers and (ii) that adequate in vitro models where the hypertrophic phenotype can be elicited and controlled are necessary to shed further light on these matter and possibly highlight new therapeutic targets.

On this regard, specifically, we added the following lines in the introduction (L77-84)

A direct causal relation between the onset of chondrocytes hypertrophic phenotype and the progression of OA has not been established yet. It is still undetermined if the hypertrophy is a trigger of OA rather than the consequence of the inflammatory and degradative environment. Different works seem however to indicate that alterations in the composition of the calcified cartilage or even of the subchondral bone, (phenomena which are both associated to endochondral ossification reminiscent processes) might precede and cause, or at least contribute to further cartilage degradation1,2. These considerations make the case for more in depth study of the phenomenon, possibly leading to new therapeutic targets. 

Introduction, Line82-83: We will then describe the models used to study chondrocytes’ assumption of a hypertrophic phenotype, which could help in dissecting the causal relationship between hypertrophy and OA

Introduction, line 80: we changed the expression from “to suppress” to “to limit” OA progression

References

  1. Changes in the osteochondral unit during osteoarthritis: structure, function and cartilage-bone crosstalk,.S. R. Goldring, M. B. Goldring, Nat. Rev. Rheumatol. 2016, 12, 632.
  2. Anatomy and physiology of the mineralized tissues: role in the pathogenesis of osteoarthrosis, D. B. Burr, Osteoarthr. Cartil. 2004, 12, 20

  1. To what degree is the chondrocyte hypertrophy observed during OA similar to that during embryonic bone development and in the vast amount of in vitro cultures of hypertrophic chondrocytes?

Response: We are thankful to the Reviewer for the comment. Concerning the resemblance of hypertrophy in OA and endochondral ossification we tried to make matters clearer in the text (Specifically in chapter 2). However, these similarities have been reviewed previously and therefore we referenced the appropriate papers:

Jaswal, A.P.; Bandyopadhyay, A. Re-Examining Osteoarthritis Therapy from a Developmental Biologist’s Perspective. Biochemical Pharmacology 2019.

Park, S.; Bello, A.; Arai, Y.; Ahn, J.; Kim, D.; Cha, K.Y.; Baek, I.; Park, H.; Lee, S.H. Functional Duality of Chondrocyte Hypertrophy and Biomedical Application Trends in Osteoarthritis. Pharmaceutics 2021.

Ripmeester, E.G.J.; Timur, U.T.; Caron, M.M.J.; Welting, T.J.M. Recent Insights into the Contribution of the Changing Hypertrophic Chondrocyte Phenotype in the Development and Progression of Osteoarthritis. Frontiers in Bioengineering and Biotechnology 2018.

R Cancedda, F Descalzi Cancedda, P.C. Chondrocyte Differentiation. Int Rev Cytol . 1995, 159, 265–358.

Concerning the similarities of OA chondrocytes and the described in vitro models we introduced, as per the reviewer suggestion, a higher degree of abstraction and reasoning of the perks and limitations of each model. These phrases can be found directly in the text in Section 3.2

  1. Are there any parameters to judge whether a given model is a suitable model of OA chondrocyte hypertrophy? To what degree are results from any of the models translatable to OA? 

Response: We thank the Reviewer for the suggestion. The whole paragraph on in vitro models has been revised to provide deeper insights on the utility and the implications of the different discussed models. Moreover, a new subchapter i.e. 3.2.6 “Considerations on the relevance of Chondrocytes hypertrophy models in OA “ was introduced discussing their suitability and to what degree their results can be translated to OA

Unfortunately, the manuscript in its current form is a rather loose collection of various topics around chondrocyte hypertrophy, while the OA focus is often lost. The manuscript would very much benefit from a more stringent focus on OA and a higher degree of abstraction.

Response: We thank the Reviewer for the suggestion. The text has been modified as per the reviewer’s suggestion to focus more on the hypertrophy aspects.

Specific comments:

  1. Pro-hypertrophic signaling pathways: The length of description appears to be biased by the group’s research focus on BMP signaling.

PTHrP/IHH and WNT signaling should be described in similar depth. For each pathway: have interventions been tested for their potential as DMOADs?

Response: We thank the Reviewer for the suggestion; the text has been modified accordingly in section 2.1 to describe IHH, WNT and TGF-β pathways.

Models: please evaluate each model not only regarding their technical challenges but more importantly regarding their potential to model OA and translatability of results.

Response: We thank the Reviewer for the suggestion, all the models have been evaluated again regarding their potential to model OA and translatability of results.

What is wrong with our models, what should we change, given the alarmingly high rate of initially promising ‘DMOADs’ that turn out to be non-effective?

Response: The text has been included in the conclusion and future perspective section as  well as required places throughout the text as per the reviewer’s suggestion to highlight what all factors need to be assessed for making a clinically relevant model of OA.

  1. Chapter Strategies to suppress chondrocyte hypertrophy: please more strongly focus on the hypertrophy aspect.

Response: The text has been modified as per the reviewer’s suggestion to focus more on the hypertrophy aspects.

Only one of the three strategies depicted in figure 2 directly targets hypertrophy (via BMP inhibition), while the others target other typical OA drivers: inflammation and mechanical overload.

Response: We thank the reviewer for the comment. We addressed it in two ways

(i) The title was changed to “Investigation and Modelling of Chondrocytes Hypertrophic Phenotype towards new Therapeutic Strategies for Osteoarthritis” which makes it more comprehensive thus leading the reader to expect information beside hypertrophy suppression in the review.

  1. ii) The idea of Fig. 2 was actually to present different strategies and/or models to address the onset of Hypertrophy in OA that is to say methodologies to recapitulate the hypertrophic phenotype in vitro, not to focus directly therapeutic strategies. The Title of the figure caption was changed to better reflect our intentions and a new sub-chapter 3.2.6 where the figure is explained more in detail was introduced.

For each potential intervention (small molecules, VEGF sequestration (hypertrophy), hypoxia, biomaterials): how can the described insights be harnessed to potentially treat OA?

Response: We have included a few lines to highlight the prospective role of hypoxia and VEGF in the pathophysiology of OA. The text has been included as per the reviewer’s suggestion.

Reviewer 2 Report

This is a very nice review of pathways leading to chondrocyte hypertrophy. There are a few instances where the authors need to explain what they mean better or possibly misinterpret data/literature.

For example, line 42 "chondrocytes change phenotype and udergo hypertrophic differentiation and as a consequence start to proliferate" - hypertrophic chondrocytes do not proliferate, OA chondrocytes revert to an earlier differentiaiton state and the theory that is discussed by the authors states that they undergo endochondral ossification (EO), or follow a similar process, that indeed leads to proliferation, and then hypertrophy. The EO process should be explained at the start of the review, as many of the pathways and genes referred to later are important for this process.

Figure 1 - the bottom of the figure serves no purpose.

The authors state that chondrocyte hypertrophy is a pathological alteration in OA, but it could be argued that it is a consequence of reprogramming to a more juvenile state in order to initiate tissue repair - it is one of the theories in the OA field and should be discussed in this review.

Table 1 - "cartilage brakes of hypertrophic differentiation" - what is meant by that phrase?

The secretome mentioned in the table is really the degradome, fragments of degraded matrix and the enzymes that degrade it

Inhibition of several pathways metioned in the review, integrins, IHH, BMP, in developement leads to embryonic lethality and/or severe skeletal malformations. it should be emphasized in the review that these interventions have to be carefully timed, and that some of these pathways are pleiotropic and have different effects and regulatiors at different stages of development.

Spntaneous models of OA are not genetically tractable, are genetically heterogenous, and some of the animal models cannot be easily maniputated, all of which can impact on studies of therapeutic intervention and should be mentioned in the review.

Dedifferentiation of chondrocytes in 2D should be covered in more detail.

Figure 2 should include more labels and explanations to be able to appreciate what is shown on the figure and how it relates to text.

Author Response

Reviewer 2

This is a very nice review of pathways leading to chondrocyte hypertrophy. There are a few instances where the authors need to explain what they mean better or possibly misinterpret data/literature.

For example, line 42 "chondrocytes change phenotype and udergo hypertrophic differentiation and as a consequence start to proliferate" - hypertrophic chondrocytes do not proliferate, OA chondrocytes revert to an earlier differentiaiton state and the theory that is discussed by the authors states that they undergo endochondral ossification (EO), or follow a similar process, that indeed leads to proliferation, and then hypertrophy. The EO process should be explained at the start of the review, as many of the pathways and genes referred to later are important for this process.

Response: The text has been included as per the reviewer’s suggestion.

Figure 1 - the bottom of the figure serves no purpose.

Response: We would like to thank the reviewer for the comment, we would like to mention that we have used this figure simply to summarise the concepts discussed in the manuscript..

The authors state that chondrocyte hypertrophy is a pathological alteration in OA, but it could be argued that it is a consequence of reprogramming to a more juvenile state in order to initiate tissue repair - it is one of the theories in the OA field and should be discussed in this review.

Response: We thank the reviewer for the suggestion. We agree with the comment and have revised the abstract and the introduction rephrasing them to highlight that there is presently no consensus on whether chondrocytes hypertrophy precedes and causes other OA hallmarks, it is a consequence of the degradative processes, or if it is an attempt at a spontaneous repair strategy. In particular the following lines were added to the text to highlight that appropriate models are needed also to shed light on these matters:

L72-83: A direct causal relation between the onset of chondrocytes hypertrophic phenotype and the progression of OA has not been established yet. It is still undetermined if the hypertrophy is a trigger of OA rather than the consequence of the inflammatory and degradative environment. Different works seem however to indicate that alterations in the composition of the calcified cartilage or even of the subchondral bone, (phenomena which are both associated to endochondral ossification reminiscent processes) might precede and cause, or at least contribute to further cartilage degradation 1,2. These considerations make the case for more in depth study of the phenomenon, possibly leading to new therapeutic targets. 

In the current review we will first summarize the cellular signaling pathways correlated to hypertrophy in OA. We will then describe the models used to study chondrocytes’ assumption of a hypertrophic phenotype, which could help in dissecting the causal relationship between hypertrophy and OA

Table 1 - "cartilage brakes of hypertrophic differentiation" - what is meant by that phrase?

Response: Table 1 has been modified as per the reviewer’s suggestion.

The secretome mentioned in the table is really the degradome, fragments of degraded matrix and the

Response: Table 1 has been modified as per the reviewer’s suggestion.

Inhibition of several pathways metioned in the review, integrins, IHH, BMP, in developement leads to embryonic lethality and/or severe skeletal malformations. it should be emphasized in the review that these interventions have to be carefully timed, and that some of these pathways are pleiotropic and have different effects and regulatiors at different stages of development.

Response: The text has been included as per the reviewer’s suggestion.

Spntaneous models of OA are not genetically tractable, are genetically heterogenous, and some of the animal models cannot be easily maniputated, all of which can impact on studies of therapeutic intervention and should be mentioned in the review.

Response: The text has been included as per the reviewer’s suggestion in section 3.1.

Dedifferentiation of chondrocytes in 2D should be covered in more detail.

Response: The text has been modified as per the reviewer’s suggestion.

Figure 2 should include more labels and explanations to be able to appreciate what is shown on the figure and how it relates to text.

Response: More labels have been included in Figure 2 as per the reviewer’s suggestion.

Reviewer 3 Report

General: This is an interesting review from a laboratory that has made important contributions to our understand of chondrocyte phenotype and plasticity during health and disease. I have made suggestions that address and correct the conceptual basis for some of the statements and for condensing some sections for more concise presentation of the major points of the review. Overall, this would be more useful to the field if it can be revised to highlight the historical contributions versus the current state-of-art knowledge can be used to drive new therapeutic strategies for OA targeting chondrocytes with hypertrophic phenotype.

Specific:

1.     It is suggested that the title be modified slightly to reflect that the target for therapy is the ‘hypertrophic chondrocyte phenotype’, which involves more than one process or marker. Tis is because “hypertrophy” really means cellular swelling or enlargement, which clearly occurs during endochondral ossification, but is poorly evident in OA.

2.     The Abstract needs deep editing for errors for missing articles and other modifiers. It seems that it was hastily written, as the body of the manuscript is somewhat better.

3.     Lines 28-31: Do you really mean “suppression of the chondrocyte” here? The use of the terminology “chondrocyte hypertrophy” (see comment 1 above) or “OA hypertrophy” is incorrect. I think what you wish to propose is that targeting features of the hypertrophic phenotype that occur in cartilage during the development of OA is a potential therapeutic approach.

4.     Lines 99-101: This statement is incorrect. Fibrillation is a macroscopic and microscopic feature that occurs on the cartilage surface as a result of biomechanical damage of the ECM proteins, including collagen and proteoglycans that were synthesized and incorporated into the matrix when the cartilage was formed. Rather, the perturbed chondrocytes secrete enzymes that degrade these matrix proteins.

5.     Lines 101-104: This misrepresents a lot of work that has gone into characterizing the chondrocyte phenotypes in different zones of cartilage and how they change during OA. In fact, most if not all of the hypertrophy-like changes occur in the deep zone with the consequence that the calcified cartilage zone expands with mineral that is somewhat different than that found in bone.

6.     Lines 105-107: The first part of this sentence should be separated to clarify where apoptosis is occurring and what are the results, e.g., chondral versus subchondral. It is not correct that apoptosis is “followed by” subchondral bone scelerosis and osteophyte formation. In fact, these changes are separate but parallel processes to those occurring in the cartilage and calcified cartilage. There is recent literature about cross talk between cartilage and bone, some of which are in the reference list but not cited here.

7.     Lines 112-113: This statement suggests that the loss of the quiescent state of articular chondrocytes and initiation of hypertrophic differentiation is interdependent.

8.     Lines 134-135: This sentence is confusing in that this paragraph is attempting to compare aspects of chondrocyte hypertrophy in development with that in OA cartilage. What is missing is a consideration of the zonal distribution of these markers in the articular cartilage in health and disease. Neither the “development of bone spicules” (not sure what this refers to but assume it occurs in the bone) nor “osteophyte formation” (involving endochondral ossification) occurs within the cartilage.

9.     In Section 2.1, summarizing candidate signaling pathways and inhibitors or activators that have been tested in vitro or in animal models, most of the reference (also cited in Table 2) are old, perhaps highlighting that attempts to develop these agents for OA therapy have been unsuccessful.

10.  Section 3 goes on for several pages with very detailed descriptions of not only in vivo models used to study OA initiation and development, but also ex vivo and in vitro models used to study chondrocyte differentiation, chondrogenesis, and cartilage regeneration and repair. In the in vivo models, there is little information about whether they can be used to study hypertrophy-like features in relation to OA. In the ex vivo and in vitro models, the hypertrophic features can be regulated, but the clinical impact is unclear. The authors might consider significantly condensing this section to highlight the most informative models. If this is part of a thesis, it might be placed in an appendix.

11.  Section 4 addresses the main points suggested by the title, but it is difficult to sort out which are the novel take-away points. Although this section cites less historical references than earlier, it is difficult to see priorities to be followed toward clinical translation.  

12.  The Conclusions and Future Perspectives section has a lot of information that might be better used to set the stage in the beginning (where the sections on endochondral ossification in development versus OA might be condensed). In the Conclusion, I would like to see more interpretive analysis of how current knowledge could lead to novel therapeutic strategies. For example, there are recent studies evaluating chondrocyte subpopulations by single cell RNAseq that might be cited to support the changes in phenotype observed in non-OA versus OA cartilage to inform on choices of targets for therapy.

Minor:

1.     Lines 32, 83: It is ambiguous to use “developed/proposed”; this would be better written as “developed or proposed’—or be more specific about the therapeutic strategies you are thinking of.

2.     Starting in the second paragraph of the Introduction (line 45), there is inconsistency in the use of nomenclature for the markers of hypertrophy and other processes. It is important to indicate whether you are talking about gene expression or protein levels in these processes (or activities in the case of proteinases). In this first sentence the abbreviations for proteins are defined, e.g., MMP-13, COLX, ALP), but later the gene symbols are mixed with protein designations. A rule of thumb to avoid confusion would be to consider “expression” as gene expression and use gene symbols.

a.     Line 114: COL10A1, MMP-13, IHH, where MMP13 should be the gene symbol

b.     Line 122/123: COLX and RUNX2

c.     Line130/131: lubricin, Aggrecan (ACAN), Collagen type II (COLII), where PRG4, ACAN, COL2A1 should be listed as gene symbols.

3.     Similarly, Table 1 needs some work to distinguish techniques that measure protein and those that measure mRNA. I would suggest not abbreviating the protein names. In fact, qRT-PCR was not available at the times of publication of some of the references indicated. In situ hybridization was used to characterize gene expression in embryonic and post-natal growth plates and in articular cartilage. More recently single cell RNAseq has been used to characterize chondrocyte subpopulations in OA cartilage, for example. I suggest rethinking the information you want to include in this table.

4.     Line 79: Please replace “flourishment”, which is not a word.

5.     Line 211: Please replace with “therapeutic inhibition or activation”.

Author Response

Reviewer 3

General: This is an interesting review from a laboratory that has made
important contributions to our understand of chondrocyte phenotype and
plasticity during health and disease. I have made suggestions that
address and correct the conceptual basis for some of the statements and
for condensing some sections for more concise presentation of the major
points of the review.

Overall, this would be more useful to the field if
it can be revised to highlight the historical contributions versus the
current state-of-art knowledge can be used to drive new therapeutic
strategies for OA targeting chondrocytes with hypertrophic phenotype.

Specific:

1.     It is suggested that the title be modified slightly to reflect
that the target for therapy is the ‘hypertrophic chondrocyte phenotype’,
which involves more than one process or marker. Tis is because
“hypertrophy” really means cellular swelling or enlargement, which
clearly occurs during endochondral ossification, but is poorly evident
in OA.

Response: the title has been modified as it follows:

“Investigation and Modelling of Chondrocytes Hypertrophic Phenotype towards new Therapeutic Strategies for Osteoarthritis”

Thus taking into account the reviewer suggestion, putting more focus on the models part, and using a more caution phrasing on the opportunity of using Hypertrophy suppression as a therapeutic approach

  1.    The Abstract needs deep editing for errors for missing articles
    and other modifiers. It seems that it was hastily written, as the body
    of the manuscript is somewhat better.

Response: The abstract has been modified as per the reviewer’s suggestion.

3.     Lines 28-31: Do you really mean “suppression of the chondrocyte”
here? The use of the terminology “chondrocyte hypertrophy” (see comment
1 above) or “OA hypertrophy” is incorrect. I think what you wish to
propose is that targeting features of the hypertrophic phenotype that
occur in cartilage during the development of OA is a potential
therapeutic approach.

Response: The text has been modified as per the reviewer’s suggestion

4.     Lines 99-101: This statement is incorrect. Fibrillation is a
macroscopic and microscopic feature that occurs on the cartilage surface
as a result of biomechanical damage of the ECM proteins, including
collagen and proteoglycans that were synthesized and incorporated into
the matrix when the cartilage was formed. Rather, the perturbed
chondrocytes secrete enzymes that degrade these matrix proteins.

Response: The text has been modified as per the reviewer’s suggestion.

  1.    Lines 101-104: This misrepresents a lot of work that has gone
    into characterizing the chondrocyte phenotypes in different zones of
    cartilage and how they change during OA. In fact, most if not all of the
    hypertrophy-like changes occur in the deep zone with the consequence
    that the calcified cartilage zone expands with mineral that is somewhat
    different than that found in bone.

Response: The text has been modified as per the reviewer’s suggestion.

  1.    Lines 105-107: The first part of this sentence should be
    separated to clarify where apoptosis is occurring and what are the
    results, e.g., chondral versus subchondral. It is not correct that
    apoptosis is “followed by” subchondral bone scelerosis and osteophyte
    formation. In fact, these changes are separate but parallel processes to
    those occurring in the cartilage and calcified cartilage. There is
    recent literature about cross talk between cartilage and bone, some of
    which are in the reference list but not cited here.

Response: The text has been modified as per the reviewer’s suggestion.

  1.    Lines 112-113: This statement suggests that the loss of the
    quiescent state of articular chondrocytes and initiation of hypertrophic
    differentiation is interdependent.

Response: The text has been modified as per the reviewer’s suggestion.

  1.    Lines 134-135: This sentence is confusing in that this paragraph
    is attempting to compare aspects of chondrocyte hypertrophy in
    development with that in OA cartilage. What is missing is a
    consideration of the zonal distribution of these markers in the
    articular cartilage in health and disease. Neither the “development of
    bone spicules” (not sure what this refers to but assume it occurs in the
    bone) nor “osteophyte formation” (involving endochondral ossification)
    occurs within the cartilage.

Response: The text has been modified as per the reviewer’s suggestion.

  1.    In Section 2.1, summarizing candidate signaling pathways and
    inhibitors or activators that have been tested in vitro or in animal
    models, most of the reference (also cited in Table 2) are old, perhaps
    highlighting that attempts to develop these agents for OA therapy have
    been unsuccessful.

Response: The text has been included in Section 2.1 and Table 2 as per the reviewer’s suggestion.

  1. Section 3 goes on for several pages with very detailed descriptions
    of not only in vivo models used to study OA initiation and development,
    but also ex vivo and in vitro models used to study chondrocyte
    differentiation, chondrogenesis, and cartilage regeneration and repair.

In the in vivo models, there is little information about whether they
can be used to study hypertrophy-like features in relation to OA. In the
ex vivo and in vitro models, the hypertrophic features can be regulated,
but the clinical impact is unclear. The authors might consider
significantly condensing this section to highlight the most informative
models. If this is part of a thesis, it might be placed in an appendix.

Response: The section 3.1 has been modified to include the information suggested by the reviewer.

11.  Section 4 addresses the main points suggested by the title, but it
is difficult to sort out which are the novel take-away points. Although
this section cites less historical references than earlier, it is
difficult to see priorities to be followed toward clinical translation.

Response: The section was used to highlight the biomaterial based drug delivery systems that have been used to modulate chondrocyte hypertrophy and/or to deduce molecular mechanisms of OA hypertrophy like changes. The novel take away points have been highlighted as per the reviewer’s suggestion.

12.  The Conclusions and Future Perspectives section has a lot of
information that might be better used to set the stage in the beginning
(where the sections on endochondral ossification in development versus
OA might be condensed).

In the Conclusion, I would like to see more
interpretive analysis of how current knowledge could lead to novel
therapeutic strategies. For example, there are recent studies evaluating
chondrocyte subpopulations by single cell RNAseq that might be cited to
support the changes in phenotype observed in non-OA versus OA cartilage
to inform on choices of targets for therapy.

Response: Conclusion has been modified to include reviewer’s suggestion.

Minor:

1.     Lines 32, 83: It is ambiguous to use “developed/proposed”; this
would be better written as “developed or proposed’—or be more specific
about the therapeutic strategies you are thinking of.

Response: “developed/proposed” has been changed to ‘Developed’

2.     Starting in the second paragraph of the Introduction (line 45),
there is inconsistency in the use of nomenclature for the markers of
hypertrophy and other processes. It is important to indicate whether you
are talking about gene expression or protein levels in these processes
(or activities in the case of proteinases). In this first sentence the
abbreviations for proteins are defined, e.g., MMP-13, COLX, ALP), but
later the gene symbols are mixed with protein designations. A rule of
thumb to avoid confusion would be to consider “expression” as gene
expression and use gene symbols.

a.     Line 114: COL10A1, MMP-13, IHH, where MMP13 should be the gene symbol

b.     Line 122/123: COLX and RUNX2

c.     Line130/131: lubricin, Aggrecan (ACAN), Collagen type II (COLII),
where PRG4, ACAN, COL2A1 should be listed as gene symbols.

Response: The gene and protein symbols have been corrected accordingly.

3.     Similarly, Table 1 needs some work to distinguish techniques that
measure protein and those that measure mRNA. I would suggest not
abbreviating the protein names. In fact, qRT-PCR was not available at
the times of publication of some of the references indicated. In situ
hybridization was used to characterize gene expression in embryonic and
post-natal growth plates and in articular cartilage. More recently
single cell RNAseq has been used to characterize chondrocyte
subpopulations in OA cartilage, for example. I suggest rethinking the
information you want to include in this table.

Response: Table 1 has been corrected accordingly.

4.     Line 79: Please replace “flourishment”, which is not a word.

Response: The word has been replaced.

  1.    Line 211: Please replace with “therapeutic inhibition or activation”.

Response: The sentence has been corrected

References

  1. Changes in the osteochondral unit during osteoarthritis: structure, function and cartilage-bone crosstalk,.S. R. Goldring, M. B. Goldring, Nat. Rev. Rheumatol. 2016, 12, 632.
  2. Anatomy and physiology of the mineralized tissues: role in the pathogenesis of osteoarthrosis, D. B. Burr, Osteoarthr. Cartil. 2004, 12, 20

Round 2

Reviewer 1 Report

Unfortunately, the manuscript has not gained considerable maturity during revision. Corrections appear to have been done hastily without proper consideration and revision. Since potential DMOADs targeting chondrocyte hypertrophy appear promising enough to have made it into phase III clinical trials, authors still cannot find a convincing argument why chondrocyte hypertrophy might indeed be a driver of OA. Authors have the tendency to inappropriately generalise findings from a specific model (e.g. mechanisms driving chondrocyte hypertrophy found during MSC in vitro chondrogenesis) to also occur during OA.

Specific remarks:

1.       Extensive language editing is required.

2.       The new title is rather unclear

3.       Line 18: While a multitude of genetic abnormalities and SNPs have been found to increase susceptibility of developing OA, I am not aware that chondrocytes change their genotype upon onset of OA (authors probably mean gene expression?).

4.       Line 28: What is ‘OA hypertrophy cellular signaling’ supposed to mean?

5.       Line 81-82: ‘we will first summarize the cellular signaling pathways correlated to hypertrophy in OA’. This would indeed be preferable, instead the authors mostly summarize pathways correlated to chondrocyte hypertrophy without detailing to what extent these mechanisms have also been observed during OA in patient samples.

6.       Table 1: MMP-1, -3 -13; MMP1, MMP2, MMP13. There appears to be a mix-up between MMP2 and MMP3?

7.       Line 155: please explain the relationship between Sik3 and Ihh.

8.       Line 157: IHH feeds back with PTHrP not with the PTH receptor.

9.       Line 158-160: ‘The PTHrP pathway ameliorates the OA development by inhibiting hypertrophic differentiation of chondrocytes mainly through the expression of the chondrocyte hypertrophy regulatory switch BAPX1/NKX3.2[80].’ The cited study described this mechanism during development. Has this been shown for human OA?

10.   Line 169-170: What specifically is meant by ‘chondrogenic characteristics’?

11.   Line 178-180: While collagen X and ALP activity have been proven to occur in human OA chondrocytes, is there convincing data on WNT-induced RUNX2 expression also in human OA?

12.   Line 189: Chondrogenesis means the process of a progenitor or stem cell differentiating into a chondrocyte – so per definition it would be inappropriate to generalize mechanisms observed in chondrocytes to also occur during the process of chondrogenesis which would occur before a cell has become a chondrocyte.

13.   Line189-190: Melatonin can act paracrine, but also intracrine and autocrine via several different pathways, mostly via PKC, PLCb, and PKA. It is not a classical WNT agonist and conclusions on the role of WNT signalling should be drawn with more care from a study working with melatonin.

14.   Line 201: please check spelling of DMOAD

15.   Line 202: also a COX2 inhibitor is not a direct WNT inhibitor, so conclusions concerning the role of WNT activity should be drawn with more care.

16.   Line 205-206: DKK1 is not an intracellular inhibitor but a secreted molecule.

17.   Paragraph 2.1.2. Wnt Pathway: would lorecivivint be worth mentioning?

18.   Paragraph 2.1.3. TGF beta pathway: TGFb is actually ascribed a dual role. While here TGFb inhibition was focused, there are other studies trying TGFb overexpressing cells as potential OA treatment.

19.   Line 217-218: ‘The increased TGF-β levels in OA joints lead to an altered differentiation of cells towards transient cartilage’. Has this been demonstrated, or is this pure speculation?

20.   Line 222 following: TGFb has a well-documented anti-hypertrophic activity. When discontinued after chondro-induction of MSCs in vitro, ALP activity rises strongly. Also, the cited study shows that under identical conditions, TGFb does not induce hypertrophy in articular chondrocytes but only in differentiating MSCs, indicating that TGFb is not per se pro-hypertrophic.

21.   Line 305: spell check ‘farmer’

22.   Line 332-333: ‘the molecular events appears during pathogenesis of osteoarthritis, are nearest to 332 those observed in patients’ sentence is unclear.

23.   Line 335: ‘variation in reproducibility of results’ wouldn’t results be either variable or reproducible?

24.   Line 432-433: ‘The degree […] should however considered carefully and its limitations accounted for while drawing conclusions with clinical relevance.’ Please revise.

Reviewer 3 Report

The authors have done an extensive job of revising the manuscript. Thank you for attending to my suggestions. I would only suggest that the use of the phrase OA "chondrocytes hypertrophic phenotype" in the title and throughout the manuscript is not good English. It would read better as "hypertrophic chondrocyte phenotype".  Or depending upon what you are trying to emphasize in the text, you could have variations such as "hypertrophic phenotype of OA chondrocytes" or other variations if you are describing certain features.